# Treatment with tumor-treating fields (TTFields) suppresses intercellular tunneling nanotube formation in vitro and upregulates immuno-oncologic biomarkers in vivo in malignant mesothelioma

Akshat Sarkari[1], Sophie Korenfeld[1], Karina Deniz[1], Katherine Ladner[1], Phillip Wong[1], Sanyukta Padmanabhan[1], Rachel I Vogel[2], Laura A Sherer[3], Naomi Courtemanche[3], Clifford Steer[3,4], Kerem Wainer-Katsir[5], Emil Lou[1,6]*

[1]Department of Medicine, Division of Hematology, Oncology and Transplantation, University of Minnesota, Minneapolis, United States; [2]Department of Obstetrics, Gynecology and Women's Health, University of Minnesota, Minneapolis, United States; [3]Department of Genetics, Cell Biology and Development, University of Minnesota, Minneapolis, United States; [4]Department of Medicine, Division of Gastroenterology, Hepatology and Nutrition, University of Minnesota, Minneapolis, United States; [5]Novocure Ltd, Topaz Building, MATAM Center, Haifa, Israel; [6]Graduate Faculty, Integrative Biology and Physiology Department, University of Minnesota, Minneapolis, United States

*For correspondence:
emil-lou@umn.edu

**Abstract** Disruption of intercellular communication within tumors is emerging as a novel potential strategy for cancer-directed therapy. Tumor-Treating Fields (TTFields) therapy is a treatment modality that has itself emerged over the past decade in active clinical use for patients with glioblastoma and malignant mesothelioma, based on the principle of using low-intensity alternating electric fields to disrupt microtubules in cancer cells undergoing mitosis. There is a need to identify other cellular and molecular effects of this treatment approach that could explain reported increased overall survival when TTFields are added to standard systemic agents. Tunneling nanotube (TNTs) are cell-contact-dependent filamentous-actin-based cellular protrusions that can connect two or more cells at long-range. They are upregulated in cancer, facilitating cell growth, differentiation, and in the case of invasive cancer phenotypes, a more chemoresistant phenotype. To determine whether TNTs present a potential therapeutic target for TTFields, we applied TTFields to malignant pleural mesothelioma (MPM) cells forming TNTs in vitro. TTFields at 1.0 V/cm significantly suppressed TNT formation in biphasic subtype MPM, but not sarcomatoid MPM, independent of effects on cell number. TTFields did not significantly affect function of TNTs assessed by measuring intercellular transport of mitochondrial cargo via intact TNTs. We further leveraged a spatial transcriptomic approach to characterize TTFields-induced changes to molecular profiles in vivo using an animal model of MPM. We discovered TTFields induced upregulation of immuno-oncologic biomarkers with simultaneous downregulation of pathways associated with cell hyperproliferation, invasion, and other critical regulators of oncogenic growth. Several molecular classes and pathways coincide with markers that we and others have found to be differentially expressed in cancer cell TNTs, including MPM specifically. We visualized short TNTs in the dense stromatous tumor material selected as

regions of interest for spatial genomic assessment. Superimposing these regions of interest from spatial genomics over the plane of TNT clusters imaged in intact tissue is a new method that we designate Spatial Profiling of Tunneling nanoTubes (SPOTT). In sum, these results position TNTs as potential therapeutic targets for TTFields-directed cancer treatment strategies. We also identified the ability of TTFields to remodel the tumor microenvironment landscape at the molecular level, thereby presenting a potential novel strategy for converting tumors at the cellular level from 'cold' to 'hot' for potential response to immunotherapeutic drugs.

## Editor's evaluation

This study is based on the hypothesis that tumor treating fields, a form of cancer therapy that exposes tumors to alternating electrical fields, has an effect on tunneling nanotubes, fine actin rich protrusions that connect cancer cells and allow intercellular communication, contributing to the tumor microenvironment and therapeutic resistance. This is an interesting hypothesis and may be of importance.

## Introduction

Intercellular communication in the dense and highly heterogeneous tumor matrix is a critical function and hallmark of invasive cancers. Multiple forms of intercellular communication have been well documented and characterized, including gap junctions, extracellular vesicles, and signaling via diffusible growth factors, among others. In the past decade, a unique form of F-actin-based cellular protrusion known as tunneling nanotubes (TNTs) has been shown to mediate direct contact-dependent intercellular communication in many cell types, and particularly, invasive cancer phenotypes.

TNTs have the ability to physically bridge cells through a spectrum of intercellular distances, with a range of 5–10 µm at extreme short-range to 100–500 µm and even longer (*Gousset et al., 2013*; *Lou et al., 2012*; *Pasquier et al., 2012*; *Pasquier et al., 2013*; *Sartori-Rupp et al., 2019*; *Tishchenko et al., 2020*). These ultrafine structures were first characterized in 2004 in PC12 cells, a cell line derived from rat pheochromocytoma (*Rustom et al., 2004*), and are morphologically and functionally distinct from other membranous protrusions such as filopodia or lamellipodia, which aid in motility and attachment to the extracellular matrix (ECM; *Nemethova et al., 2008*). Unlike filopodia and lamellipodia, TNTs are non-adherent to the substratum in cells cultured in vitro (*Tishchenko et al., 2020*; *Rustom et al., 2004*; *Desir et al., 2018*; *Dubois et al., 2020*). TNTs have been identified in many forms of cells, including fibroblasts, epithelial cells, and neurons, but are prominently upregulated in cancer cells (*Lou et al., 2012*; *Tishchenko et al., 2020*; *Desir et al., 2018*; *Desir et al., 2016*; *Desir et al., 2019*; *Cole et al., 2021*; *Hanna et al., 2019*). The potential for a single TNT to remain stable for hours, combined with its upregulation in cancer phenotypes, indicates that TNTs may be capable of mounting a rapid communication response to external stimuli or insults, including chemotherapeutic drugs (*Tishchenko et al., 2020*; *Desir et al., 2018*). However, the mechanism(s) of TNT formation and the role of actin in TNT formation and stability across cell types remain largely unknown.

Tumor-Treating Fields (TTFields) therapy is a novel therapeutic strategy in clinical use for patients with several forms of advanced cancers, including glioblastomas and malignant pleural mesotheliomas (MPM). It is based on the principle of using low-intensity alternating electric fields to disrupt microtubules in cancer cells undergoing mitosis. These fields apply forces on charges and polarizable molecules inside and around cells. TTFields can disrupt mitosis in malignant cells due to its ability to interfere with mitotic spindle assembly through impairment of microtubule polymerization (*Giladi et al., 2015*). Microtubules are essential in ensuring that chromosomes attach and segregate correctly during metaphase and anaphase, respectively. The individual subunits of microtubules, known as tubulins, are heterodimers with two distinct protein domains, in which one has a positive end and one has a negative end, creating a dipole (*Marracino et al., 2019*). If microtubules are not allowed to polymerize, cell division cannot occur, and this is typically followed by chromosomal abnormalities and mitotic cell death (*Forth and Kapoor, 2017*). Additionally, TTFields application creates a nonuniform electric field during the telophase phase of mitosis due to alignment of the cell in cytokinesis, leading to a process known as dielectrophoresis, which can also result in improper cell division and mitotic death (*Mun et al., 2018*). Other mechanisms of action have also been demonstrated, including

downregulation of DNA damage response, impairment of cancer cell migration, and induction of anti-cancer immunity (*Porat et al., 2017*). Unlike systemic cancer therapies, TTFields delivery is focused on the tumor area, thus minimizing effects on non-malignant cells outside the treated area. Due to differences in geometrical and electrical properties, the TTFields frequency range is deleterious to cancer but not to benign cells and is optimized to a specific tumor type (*Mun et al., 2018*; *Porat et al., 2017*). This technology is currently applied concomitantly with standard-of-care treatment approaches for patients with glioblastoma and mesothelioma, with clinical trials also ongoing in many other forms of metastatic or difficult-to-treat forms of cancer.

The bulk of studies to date on cellular effects and mechanism(s) of TTFields has focused on disruption of microtubules, leading to decreased cell division. For this study we hypothesized that TTFields also affect formation of F-actin based TNTs in intact cells. We have previously reported that TNTs are significantly upregulated in multiple forms of MPM, which serves as an excellent model system for studying and characterizing cellular structure, function, and dynamics of TNT-mediated intercellular communication of cellular signals. Here, we report the effects of TTFields on TNTs connecting MPM cells in vitro, and on cell-free monomeric and filamentous actin. We also examined differential expression of gene pathways of immune response, proliferative growth, and other hallmarks of MPM in an in vivo animal model in order to elucidate the impact of TTFields on the expression of genes known to be involved in TNT formation.

## Results

### Establishing TTFields application impact on TNT formation in malignant mesothelioma cells and optimizing parameters

We utilized two mesothelioma cell lines, MSTO-211H (biphasic histologic MPM subtype) and VAMT (sarcomatoid MPM), to investigate effects of TTFields application on TNT formation and function. These two cell lines were used having previously demonstrated that they reliably and reproducibly form TNTs in culture under variable conditions and are thus ideal for in vitro studies. TTFields were applied to cells in vitro using two devices: inovitro, which applies TTFields to cells in culture to which the electrodes from the power supply provide a pre-specified level of intensity and frequency of the alternating electric fields; and inovitro Live, in which the configuration is adapted for continuous administration of TTFields while permitting time-lapse microscopic imaging. We first tested the inovitro Live device to treat MSTO-211H at differing frequencies to establish parameters used to impact TNT formation. Previously, bidirectional application of TTFields has shown increased cytotoxicity relative to unidirectional delivery (*Kirson et al., 2004*), with highest cytotoxicity for MSTO-211H cells displayed at a frequency of 150 kHz (*Mumblat et al., 2021*). Instead, we sought to elucidate the initial impact of TTFields on TNT protrusion formation, which may require differing frequencies than what is demonstrated to be most effective for a cytotoxic effect. Thus, we tested differing frequencies and directional vectors for TTFields application. TTFields intensity was administered at 1.0 V/cm but a frequency of either 200 kHz or 150 kHz was delivered bidirectionally or unidirectionally over a 72 hr period to MSTO-211H cells; these two frequencies were selected for testing because the approved devices for TTFields therapeutic delivery is applied at these frequencies (*Figure 1A*). We found that by 24 hr, unidirectional TTFields treatment at 200 kHz had fewer TNTs than the control (p=0.004) and bidirectional application at 150 kHz (p=0.005). As compared to control, bidirectional application at 200 kHz also had statistically significantly fewer TNTs (p<0.0001). However, unidirectional application at 150 kHz resulted in no significant differences in TNT formation throughout the 72-hr period (*Supplementary file 1*). At times 48 and 72 hours, we also observed the decline in TNTs. As with previous studies, once cells become densely packed, they form fewer TNTs (*Ady et al., 2014*). Together the data indicated that applying TTFields at 200 kHz unidirectionally is more effective at decreasing TNT formation in MSTO-211H cells and we utilized this frequency for the rest of our experiments.

### TTFields treatment suppresses formation of TNTs between biphasic malignant mesothelioma cells when applied at 200 kHz and 1.0 V/cm

Next, we tested the ability of applied TTFields using the inovitro device to treat MSTO-211H and VAMT cells independently plated on treated coverslips using a low intensity of 0.5 V/cm TTFields treatment over a 72 hr period. TTFields did not significantly alter the number of TNTs or cells at

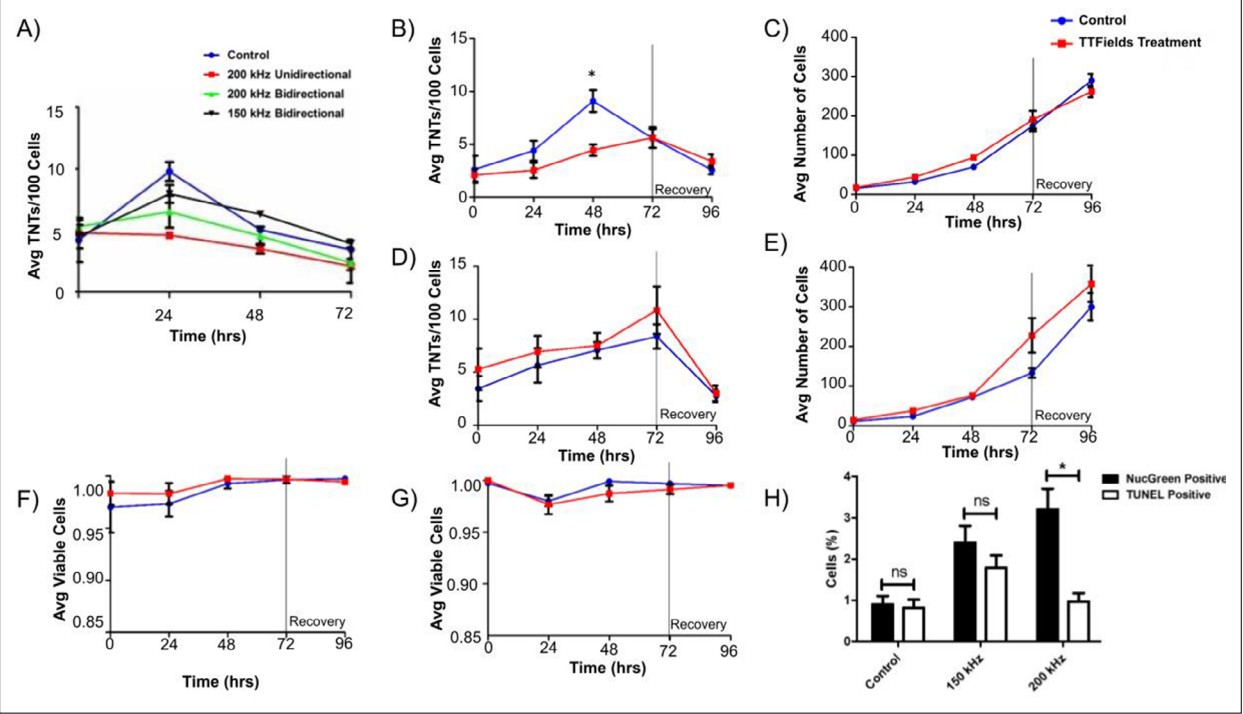

**Figure 1.** TNT formation, cell growth, and cell viability of MSTO-211H and VAMT malignant mesothelioma cells. (**A**) TNT formation in MSTO-211H following continuous TTFields exposure at 1.0 V/cm while varying frequency and field direction. 40,000 MSTO-211H cells were plated in a 35 mm dish and exposed to TTFields treatment at 1.0 V/cm with the above varying parameters; media was changed every 24 hours. Additional data for 150 kHz Unidirectional treatment are available in *Figure 1—figure supplement 1*. (**B–C**) TNT formation and cell growth in MSTO-211H following TTFields exposure when compared to control. As above, 40,000 cells were plated and were exposed continuously to TTFields bidirectionally; at 72 hr, TTFields treatment was discontinued to assess recovery of TNT formation (n=3). (**D–E**) TNT formation and cell growth in VAMT following TTFields exposure with methodology as listed in B-C (n=3). (**F–G**) Cell viability in both MSTO-211H (**F**) and VAMT (**G**) respectively following TTFields exposure. Cell viability and cytotoxicity was measured through NucGreen Dead 488 expression, which assesses loss of plasma membrane integrity. Seven random fields of view were selected and the ratio of live:dead cells was recorded (n=3). (**H**) Cell viability measured by TUNEL assay and NucGreen Dead 488 expression in MSTO-211H exposed to TTFields at 150 and 200 kHz. MSTO-211H cells were treated with TTFields for 48 hr at either 150 kHz or 200 kHz. At the 48 hr time point, cell viability was measured through the TUNEL assay or through measuring fluorescent expression of Nuc Green Dead 488. The percentage of nonviable cells was graphed as compared to a control. A representative image of TUNEL positive control is displayed in *Figure 1— figure supplement 2*. Statistical significance was assessed as a result of three independent experiments, with a linear mixed model used in A and heteroscedastic t-test used in B.

The online version of this article includes the following figure supplement(s) for figure 1:

**Figure supplement 1.** TNT formation in MSTO-211H following TTFields unidirectional delivery at 150 kHz (1 V/cm).

**Figure supplement 2.** Representative images of the TUNEL assay in MSTO-211H.

**Figure supplement 3.** TTFields delivered at low intensity (0.5 V/cm) have no effect on TNT formation or on cell proliferation in MSTO-211H or VAMT mesothelioma cells.

**Figure supplement 4.** TNT formation and cell growth in MSTO-211H after 48 hr of TTFields exposure (1 V/cm, 400 kHz).

**Figure supplement 5.** Cell count in MSTO-211H at 1 V/cm, 200 kHz, and seeded at 10,000 cells.

this intensity in either cell line (*Figure 1—figure supplement 3*). Once our experimental set-up was calibrated, we assessed effects of TTFields applied at a more standard intensity of 1.0 V/cm with a 200 kHz frequency bidirectionally to evaluate the potential impact on TNT formation. Although we demonstrated the highest impact on TNT formation with unidirectional fields, we desired to emulate clinical conditions and efficacy as closely as possible and thus utilized bidirectional electric fields. Both cell lines were treated with TTFields over a 72 hr period to assess TNT formation and cell growth, with further assessment for an additional 24 hr after TTFields was discontinued to observe any latent effect or recovery of TNT formation. Over the 72 hr treatment period, we noted a statistically significant difference in TNT formation at 48 hr with MSTO-211H cells, but this difference was not present at 72 hr (*Figure 1B*, p=0.018). Additionally, over the 24 hr following treatment stoppage, TNT formation

decreased further in both the control and treatment groups and cell density continued to increase (*Figure 1B and C*). In fact, cell growth increased steadily in both treatment and control groups at nearly exponential rates, to reach confluency by the end of the experiments, indicating there was no latent effect on either TNT formation or cell growth from TTFields application. Unlike MSTO-211H, when VAMT cells were subjected to TTFields at 1.0 V/cm, no significant differences were seen between treatment and control groups in either TNT formation or cell growth (*Figure 1D and E*). Lastly, we examined TNT formation in MSTO-211H following TTFields exposure at peak frequency threshold of 400 kHz to assess TNT suppression at a frequency substantially higher than what is used clinically (*Figure 1—figure supplement 4*). We measured TNT formation and cell growth after 48 hr of treatment based on our findings of maximal TNT suppression at that time point. No significant differences in either TNT formation or cell growth were observed.

## Assessment of cell viability and DNA fragmentation following TTFields treatment

With TTFields application at 1 V/cm, a cytotoxic effect on cells was expected. However, as reported above, both MSTO-211H and VAMT continued to divide, even when monitored 24 hr following treatment. To confirm that the cells were indeed viable, we next performed cell viability assays at all time points on randomly selected fields of view using NucGreen Dead 480. In all cases, cell viability of both control and treatment groups was >95% (*Figure 1F and G*), demonstrating no induction of cell death in the treated cells. Because TTFields exposure is known to affect cancer progression, we measured DNA fragmentation through the TUNEL assay. To confirm our earlier results with NucGreen Dead 480 and investigate cell viability at 150 kHz, we also repeated cell viability assays with NucGreen Dead 480 at both 150 kHz and 200 kHz at 1.0 V/cm. Knowing that maximum TNT suppression occurred in MSTO-211H at 48 hr, we performed both assays at the 48 hr time point. For both TUNEL and NucGreen Dead assays, we noted minimal cell death with 2.4% and 1.8% mean cell death respectively at 150 kHz and 3.2% and 1% mean cell death at 200 kHz when referenced with a negative and positive control (*Figure 1H*). As our findings of exponential cell growth and low cell death are in contrast to previous TTFields application studies, we repeated the experiments above at 1.0 V/cm and 200 kHz, but this time plated cells at a much lower density. In concurrence with others, we noted an 80% reduction in cell count in the TTFields-treated group when compared to control by the 72–96 hour time point (*Figure 1—figure supplement 5*, p=0.003).

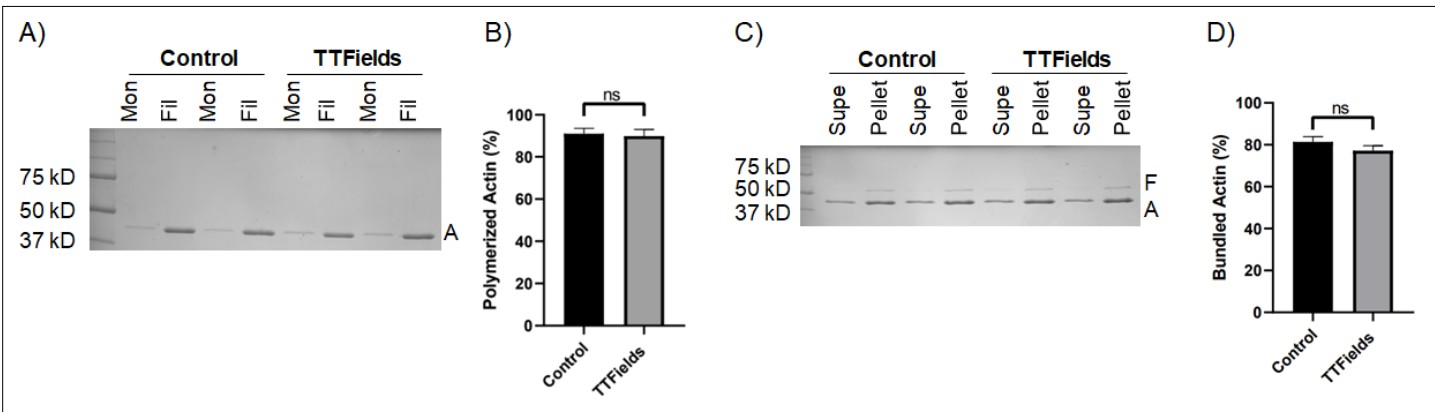

**Figure 2.** The effect of TTFields application on actin polymerization and actin filament bundling. (**A**, **B**) Sedimentation assays quantifying actin polymerization. Purified actin monomers were polymerized for 1 hr with TTFields (200 kHz, 1.0 V/cm, 37 °C) and without TTFields (37 °C) treatment. Reactions were centrifuged at 100,000 x *g* to pellet filamentous actin and analyzed by SDS-PAGE. Mon refers to monomeric actin (supernatant), Fil refers to filamentous actin (pellet). A indicates the actin protein band (42 kDa). (**C**, **D**) Co-sedimentation assays quantifying bundling of actin filaments by the bundling protein fascin. Pre-polymerized actin filaments were incubated with fascin for 1 hr with TTFields (200 kHz, 1.0 V/cm, 37 °C) and without TTFields (37 °C) treatment. Reactions were spun at low-speed (10,000 x *g*) to pellet bundles and analyzed by SDS-PAGE. The supernatant contains monomeric actin and individual filaments. The pellet contains bundled actin. F, A indicate fascin (55 kDa) and actin (42 kDa) protein bands. The gels (**A**, **C**) represent one representative experiment. The graphs (**B**, **D**) represent the average of three experiments, and the error bars are the standard deviation.

The online version of this article includes the following source data for figure 2:

**Source data 1.** Source data for results shown in *Figure 2*, titled 'The effect of TTFields application on actin polymerization and actin filament bundling'.

## Effect of TTFields exposure on actin polymerization and filament bundling

There are many unidentified molecular factors in the actin polymerization mechanism that form TNTs, including actin nucleators, elongators, bundlers, and destabilizers. In addition, there are membrane bound proteins involved in the process, and some of these components may differ between cell types. Filamentous actin forms the structural basis of the interior of TNTs. Because we observed a reduction in MSTO-211H TNTs with TTFields at 1.0 V/cm, and noting that tubulin depolymerization and polymerization has been observed to be directly impacted by TTFields treatment (*Giladi et al., 2015*), we next sought to determine what effects TTFields might have directly on actin at the polymer level. To accomplish this, we performed actin sedimentation experiments to examine both polymerization and bundling. Actin monomers in solution were combined with a KCl, MgCl$_2$, and EGTA containing buffer to initiate polymerization, and for experimental samples, treated with TTFields at 1.0 V/cm-200 kHz with the inovitro device. After one hour of incubation, solutions were spun down and run on an SDS-PAGE. Surprisingly, there was no difference between samples treated with or without TTFields (*Figure 2A and B*; *Figure 2—source data 1*). For both control and treated samples, actin was predominately found in the filamentous form. If TTFields did not directly alter actin polymerization, we considered a role for other components of the actin-based protrusion mechanism. As an initial experiment, we analyzed the actin bundling protein fascin to determine whether it was affected by TTFields. Again, there was no difference in the amount of actin bundling between TTFields-treated samples and controls, indicating that TTFields likely affect TNT formation by other factors in this system (*Figure 2C and D*; *Figure 2—source data 1*).

## The addition of chemotherapeutic agents to TTFields leads to reduced TNT formation and cell growth

TTFields are used clinically in patients concomitant with standard-of-care chemotherapy. The degree to which the interactions between and effects of TTFields and chemotherapy given together are synergistic has been shown when adding pemetrexed to cisplatin chemotherapy (*Mumblat et al., 2021*). Demonstrating that TTFields exposure suppresses TNTs in MSTO-211H cells, we leveraged our ability to assess dynamic changes over time through continuous application of TTFields while capturing live-cell reaction during time-lapse microscopy. To do this, we utilized inovitro Live, a device that applies continuous TTFields while inserted into a tissue culture plate, and which is placed in an environmentally controlled microscope chamber. This experimental arrangement permits continuous viewing, imaging, and management of cells undergoing TTFields treatment in real time. Thus, we posited that addition of standard-of-care chemotherapeutic drugs cisplatin (C) and pemetrexed (P) (Alimta) would work at least additively, and possibly synergistically, in combination with TTFields.

We performed a series of time-lapse experiments with 6 experimental groups: Control, TTFields only (1.0 V/cm, 200 kHz), Cisplatin w/o TTFields, Cisplatin + TTFields, Pemetrexed +Cisplatin w/o TTFields, and Pemetrexed +Cisplatin + TTFields (*Figure 3A and B*). When TTFields treatments were applied for 72 hr, a downward trend in TNT formation was observed compared to the control group

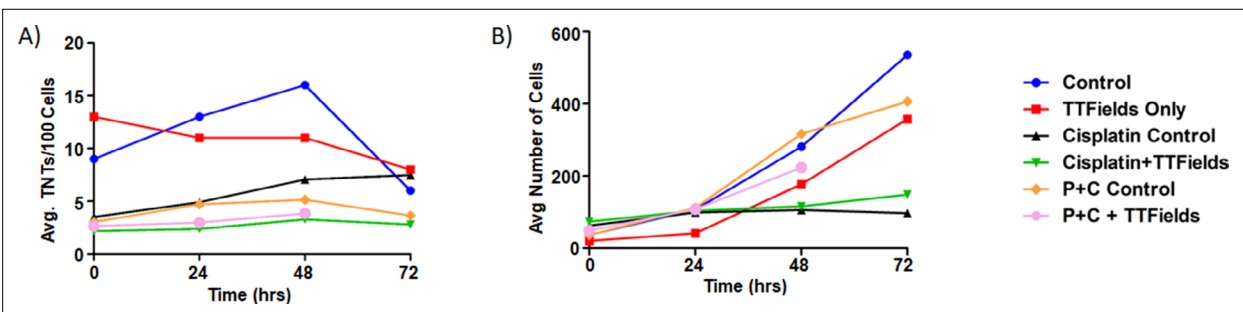

**Figure 3.** The effect of synergistic TTFields and chemotherapeutic exposure on MSTO-211H TNT formation and cell growth. (**A**) TNT formation following treatment with cisplatin and cisplatin + pemetrexed over 72 hr. Intensity and frequency were set at 1.0 V/cm and 200 kHz respectively with bidirectional field delivery. (**B**) Cell growth with chemotherapeutic reagents (C, cisplatin and P, pemetrexed) at 1.0 V/cm, 200 kHz, bidirectional. Results are indicative of one independent experiment (n=1) but with 45 technical replicates (TNTs/cell measured in multiple regions within the same experiment) averaged for each time period and condition.

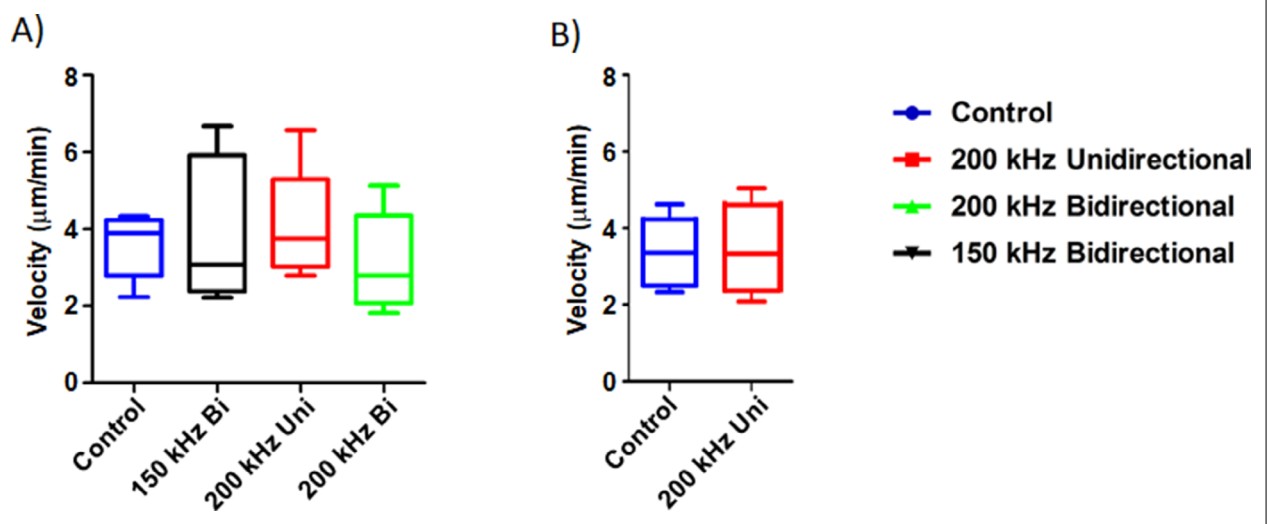

**Figure 4.** The effect of TTFields application on cargo transfer along TNTs. (**A**) Cargo velocity with 1.0 V/cm, 150 or 200 kHz, unidirectional or bidirectional TTFields application. (**B**) Mitochondrial velocity with 1.0 V/cm, 200 kHz unidirectional TTFields application. Results are indicative of three independent experiments (n=3).

(*Figure 3A*). This difference was most pronounced at the 48 hr time point, a result that was replicated from our original inovitro data in MSTO-211H (*Figure 1A*). Cell proliferation approximated an exponential growth curve for both the control and TTFields treatment groups, although the TTFields group had lower cell counts by 72 hr (*Figure 3B*). Next, we examined TNT formation and cell proliferation when the chemotherapeutic drug cisplatin was added at a physiologically relevant concentration (160 nM) to cells in culture. TNT formation was suppressed throughout the 72 hr period when compared to the control (*Figure 3A*). Cell growth was also suppressed, despite a higher cell count observed in the cisplatin group at the 0 hr time point (*Figure 3B*). When TTFields treatment at 1.0 V/cm and cisplatin were added concurrently, TNT suppression was even more pronounced, and this suppression again lasted for 72 hr. TNT formation was also suppressed in cells cultured with cisplatin and pemetrexed without TTFields at all time points when compared to the control (*Figure 3A*). Cell growth was similar to controls for 0 and 24 hr, but by 72 hr the cell growth of the cisplatin and pemetrexed treatment was suppressed (*Figure 3B*). When TTFields treatment was combined with cisplatin and pemetrexed, TNTs were also suppressed for the duration of the experiment similar to treatment with only cisplatin and pemetrexed. Cell growth under this condition was similar to controls at 0 and 24 hr, with cells in the treatment group ending at 72 hr with fewer cells than the control.

## TNT cargo transport

TNTs mediate a cell contact-dependent form of transfer of cellular contents resulting in direct communication between cells. As TTFields applied at 1 V/cm suppressed formation of TNTs in MSTO-211H, we next sought to assess the effects of TTFields at these parameters on the ability of intact TNTs to mediate intercellular transport. We sought to track two kinds of TNT cargo: gondolas (bulges) representing cellular cargo being transported via TNTs that can be tracked with brightfield microscopy, and mitochondria, which we tracked using standard commercially available fluorescent labels. Gondolas were analyzed in MSTO-211H cells treated with no TTFields (control) and 200 kHz unidirectional, 200 kHz bidirectional, and 150 kHz bidirectional TTFields (*Figure 4A*). Images were captured every 60 s for 1 hr and analyzed by the Fiji-ImageJ Manual Tracking plugin. In the control group, the average velocity of TNT transport was 3.59 μm/min. The average velocity of TNT transport was 3.94 μm/min, 4.07 μm/min, and 3.07 μm/min for cells treated with TTFields delivered unidirectionally at 200 kHz, bidirectionally at 200 kHz, or bidirectionally at 150 kHz, respectively. These findings indicated that there were no observable differences in visible cargo velocities moving through TNTs in cells treated with or without TTFields.

Transport of mitochondria through TNTs has been extensively characterized to date (*Lou et al., 2012*), and could indicate another way TTFields impact TNT functionality. MSTO-211H cells were

stained with MitoTracker Orange (Thermo Fisher Scientific) and plated for optimal TNT formation. The following day they were either treated with or without TTFields applied unidirectionally at 1.0 V/cm and 200 kHz. Images were captured every 60 s for 1 hr, and fluorescently labeled mitochondria were analyzed using Fiji-ImageJ Manual Tracking plugin. In the control group, we showed that the average velocity of mitochondria was 3.32 µm/min, with a standard deviation of 0.504 um/min, and with TTFields treatment an average velocity of 3.43 µm/min, with a standard deviation of 0.17 µm/min (*Figure 4B*). This finding indicated that similar to gondolas, there was no observable effect of TTFields on mitochondrial transfer in TNTs at the intensity and frequency that suppressed formation of TNTs.

## Spatial transcriptomic signatures of tumors treated with TTFields: Genetic effects of applying the TTFields to treat tumors in an in vivo animal model of malignant mesothelioma

At present, there is no validated specific structural biomarker for TNTs, though there are proteins known to be upregulated in TNT formation in cancer phenotypes. Approaches to molecular analysis that could uncover TNT-specific biomarkers with high sensitivity would be an important advance for the field. At the same time, there are few studies reporting alterations in molecular pathways associated with TTFields-based treatment of cells or in vivo tumor models. We thus sought to leverage a spatial genomics approach to determine whether genes that have been associated with TNT formation and maintenance, are differentially expressed in a spatially distributed manner in intact tumors; and also to identify a convergent population of genes that are both differentially expressed following treatment using TTFields and also implicated in TNT biology. Within that context, to characterize alterations induced by TTFields at the genetic and molecular levels, and potential effects in particular

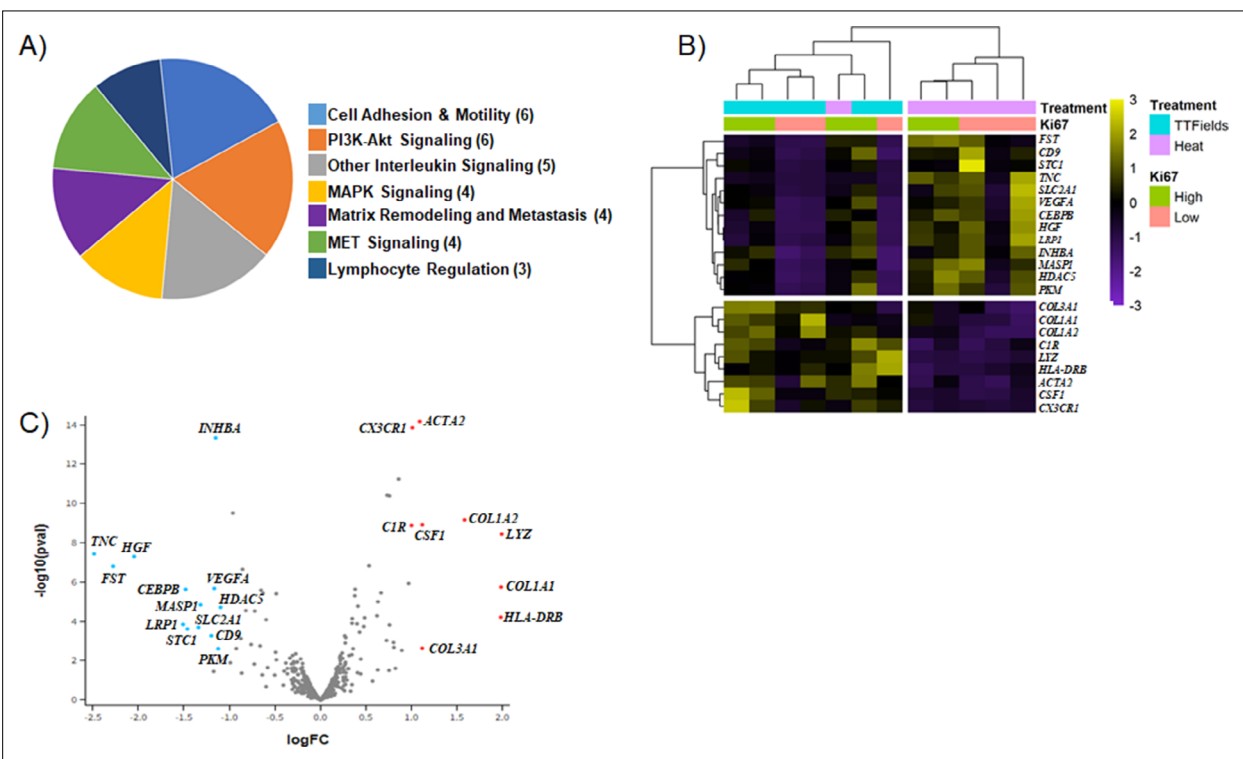

**Figure 5.** Differentially expressed genes (DEG) of TTFields-treated tumors. (**A**) Categories of genes found to be differentially expressed. () indicates the number of genes, that fall into a given category. (**B–C**) Heatmap and Volcano plot generated by spatial omics analysis.

The online version of this article includes the following source data and figure supplement(s) for figure 5:

**Source data 1.** GeoMX cancer transcriptome atlas gene panel containing corresponding protein function and tissue compartment information for genes involved in cancer progression.

**Figure supplement 1.** Cluster analysis of genes in high versus low Ki67 ROI within each treatment group.

**Table 1.** Differentially expressed genes (DEG) of TTFields-treated tumors.

| Gene | Log2 Fold Change | p-Adjusted |
| --- | --- | --- |
| LYZ | 1.9942 | 2.197E-07 |
| COL1A1 | 1.9875 | 6.490E-05 |
| HLA-DRB | 1.9839 | 1.179E-03 |
| COL1A2 | 1.5858 | 5.548E-08 |
| CSF1 | 1.1208 | 8.689E-08 |
| COL3A1 | 1.1203 | 2.727E-02 |
| ACTA2 | 1.0931 | 4.455E-12 |
| CX3CR1 | 1.0117 | 4.598E-12 |
| C1R | 1.0034 | 8.689E-08 |
| HDAC5 | 1.0960 | 4.199E-04 |
| PKM | 1.1221 | 2.744E-02 |
| INHBA | 1.1502 | 1.010E-11 |
| VEGFA | 1.1652 | 7.205E-05 |
| NR4A1 | 1.1725 | 1.958E-01 |
| CD9 | 1.1966 | 7.562E-03 |
| MASP1 | 1.3157 | 3.359E-04 |
| SLC2A1 | 1.3392 | 3.051E-03 |
| STC1 | 1.4611 | 3.623E-03 |
| CEBPB | 1.4803 | 7.251E-05 |
| LRP1 | 1.5079 | 2.348E-03 |
| HGF | 2.0459 | 2.519E-06 |
| FST | 2.2774 | 6.837E-06 |
| TNC | 2.4863 | 2.002E-06 |

on TNT-associated biomarkers, we performed spatial genomic analysis on an animal model of mesothelioma treated with TTFields, or alternately with heat as a sham for a negative control.

Eight total mice were injected with AB1 mesothelioma cells and assessed for tumor growth until they reached 200 mm$^3$ in size. Four mice each were treated with TTFields using the inovivo device, or heat sham for a negative control, as described in Materials and Methods. Following TTFields or heat application, the mice were sacrificed, and the tumors were formalin fixed and paraffin embedded. One section from all eight tumors was adhered to a glass slide, as per Nanostring GeoMx instructions, from which a total of twelve regions of interest (ROI) were chosen. Six ROIs were from TTFields-treated tumors, and six from heat sham-treated tumors. These 6 ROIs were further divided into high or low Ki67 positive regions, as a measurement of mitotic index. NanoString's GeoMx Digital Spatial Profiler system, with their mouse cancer transcriptome atlas (Catalog Number: GMX-RNA-NGS-CTA-4), was used to analyze the expression level of 1812 genes within our ROIs.

Analysis of gene expression showed that 22 of the CTA 1812 genes analyzed were differentially expressed (*Figure 5*, *Table 1*, *Figure 5—source data 1*). Broadly we found that the application of TTFields results in regulation of genes involved in cell adhesion and motility, PI3K-AKT signaling, and immune response; and to a lesser extent MAPK and MET signaling, and matrix remodeling-metastasis (*Figure 5A*). We focused on the subset of genes from the low Ki-67 ROIs, as the extent of their differentially expressed genes (DEG) was more pronounced. We reasoned that as the cells in these regions had a low rate of cell division, they were more affected by TTFields application, and thus potentially would be more likely to reveal genes that regulate TNT formation. We also performed analysis

comparing ROI from areas of high vs. low Ki-67 index, in TTFields and heat sham-treated clusters (*Figure 5—figure supplement 1*).

The genes most prominently affected (downregulated) in TTFields treated tumors as compared to the heat controls, were Tenascin C (TNC), FST, and HGF (*Figure 5B and C*, *Table 1*). TNC is a glycoprotein involved in the epithelial-to-mesenchymal transition, and was previously found by our group to be upregulated in TNT promoting conditions (*Ady et al., 2014*). TNC expression was 2.5-fold lower in TTFields-treated tumors compared to negative control (*Table 1*). In contrast, upregulation was most prominent for HLA-DRB, LYZ, COL1A1, and COL1A2. LYZ is associated with neutrophil degranulation and host defense peptides, whereas COL1A1, COL1A2 along with HGF, TNC, and VEGFA are all part of the PI3K pathway, which plays an important role in cancer progression, and has been implicated in TNT regulation (*Wang et al., 2011*). Expression of immunogenic markers with implications for efficacy of immuno-oncology therapeutic strategies were also found, and included CX3CR1, which was upregulated in TTFields-treated tumors overall, as well as the aforementioned HLA-DRB, C1R and COL3A1. Markers of angiogenic activity, such as VEGFA, which are also implicated in EMT, hypoxia signaling pathways and cell adhesion and motility, also were notably downregulated. In sum, application of TTFields altered a spectrum of metabolic and molecular signaling pathways that are well established in cell proliferation and division, ancillary pathways associated with construction and maintenance of the tumor matrix, while at the same time upregulating certain immunogenic markers.

After investigating differential gene expression in mice tumors treated with TTFields vs. control (heat sham), we imaged excised tumors using confocal fluorescence microscopy to identify and characterize TNTs and similar protrusions within the intact tumor specimens. Mice tumors were prepared as described above and sectioned into 15 µm slices, and then mounted, deparaffinized, and stained with Sytox Green 488 and Alexa Fluor 647 Phalloidin as described in Methods below. Z-stacks and images were acquired at 60 x oil immersion. In the control heat-treated native tissue, we observed TNTs, representative samples of which are shown in *Figure 6* and *Figure 6—figure supplement 1*. The one shown in *Figure 6* is in the form of a TNT protrusion in a high Ki-67 expressed ROI, 4.5 µm long in the XY plane at 60 X. Analysis of the z-stack at Nyquist sampling using IMARIS 3D revealed an additional TNT protrusion of the same length labeled by Alexa Fluor 647 phalloidin; and both were suspended three-dimensionally within the tissue microenvironment of the sample (*Figure 6—video 1*). In TTFields-treated tissue, we similarly identified TNTs, or at least TNT-like protrusions, via analysis of a z-stacked 60 x field of view with high Ki-67 expression measuring approximately 4 µm in length (*Figure 6C*; *Figure 6—video 2*). Each of the identified protrusions was consistent in appearance with our previous results identifying TNTs/TNT-like protrusions connecting cells in intact human tumor specimens from mesothelioma patients. The observed differences resulted primarily from a much more densely packed stromatous environment in this animal model. The protrusions were both shorter in comparison to the human version, and also much shorter in length than the TNTs we observed in vitro.

## Discussion

TTFields are low intensity (1–3 V/cm) alternating electric fields applied at frequencies ranging from 100 to 400 kHz (*Kirson et al., 2004*; *Mumblat et al., 2021*) and have been shown to impact polar proteins during cellular replication, specifically tubulin. Because the main component of TNTs is F-actin molecules, which have an electrochemically polar nature, we decided to study the effect of TTFields treatment on TNT formation in mesothelioma. In this study, we investigated the ability of TTFields to affect TNT formation and function in MPM, and also evaluated genetic signatures affected by TTFields treatment of MPM in an animal model. We found that TTFields significantly suppressed formation of TNTs in the biphasic (epithelioid plus sarcomatoid) form of MPM represented by the MSTO-211H cell line, when TTFields were applied at standard intensity of 1.0 V/cm, 48 hr after initiation of treatment. No significant differences were seen at 24 hr, nor subsequent to 48 hr, when cell crowding under cell culture conditions naturally leads to fewer TNTs. We found no detectable effect on TNTs with the pure sarcomatoid cell line VAMT. We assessed free actin, in monomeric and filamentous form, and found no detectable differences due to TTFields in this context. Spatial genomic assessment of intact MPM tumors following TTFields detected a notable upregulation of immuno-oncologic biomarkers, with concurrent downregulation of multiple metabolic, cell signaling, and cell growth pathways associated with dysregulation of MPM and other cancers. Some of these signals have also been implicated by our team and others in TNT activity in MPM and similar cell types.

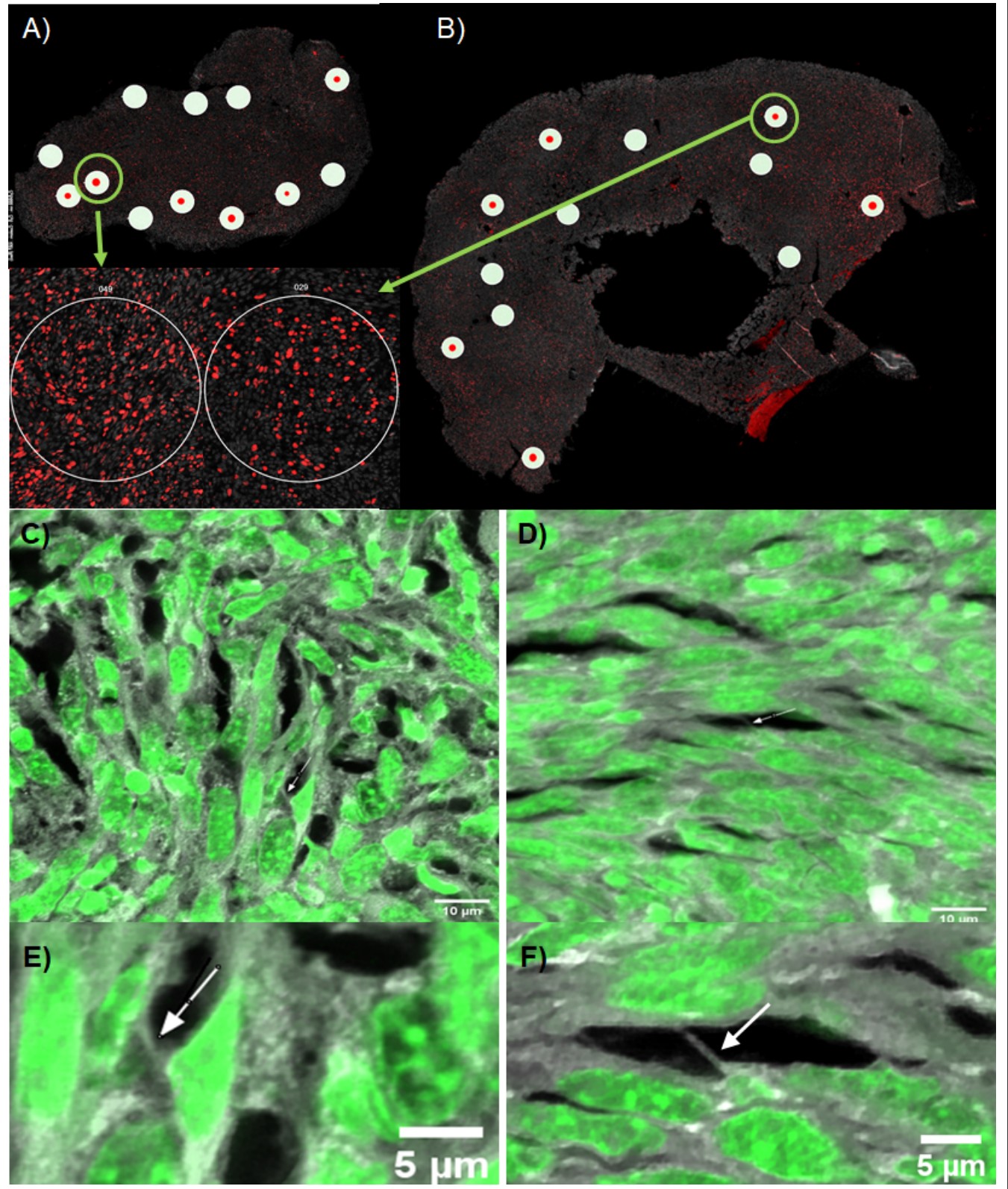

**Figure 6.** Representative fluorescence images of TNTs in intact mesothelioma tumor from our mouse model. BALB/C mice were injected with AB1 mesothelioma cells and resulting tumors were excised, sectioned, and used for NanoString GeoMx spatial profiling, as well as for imaging on a NIKON A1RSi light confocal microscope. (**A–B**) Map of tumor sections and selected ROIs used for GeoMx spatial profiling (white circles) and confocal microscopy (green rings) of (**A**) TTFields treated tumor; and (**B**) heat sham treated tumor. Insets are enlarged images of the high Ki67 ROI images

*Figure 6 continued*

selected for confocal microscopy, where Ki67 is stained in red, and nuclei in gray. (**C**–**F**) Z-stack projections of TNTs identified in selected ROIs, TTFields-treated tumor slices (**C**, **E**) and heat control slices (**D,F**) respectively. Z-stacks were acquired at 60 X using Nyquist sampling with Sytox Green 488 nuclear stain and Alexa Fluor Phalloidin 647. Arrows point to TNT protrusions within the sample.

The online version of this article includes the following video and figure supplement(s) for figure 6:

**Figure supplement 1.** Additional representative TNT protrusion in a TTFields-treated resected murine mesothelioma tumor.

**Figure 6—video 1.** 3-dimensional reconstruction of the heat sham-treated (negative control) intact tumor depicted in *Figure 6A*.

https://elifesciences.org/articles/85383/figures#fig6video1

**Figure 6—video 2.** 3-dimensional reconstruction of the TTFields-treated MPM intact tumor depicted in *Figure 6C*.

https://elifesciences.org/articles/85383/figures#fig6video2

Previous studies of TNTs have demonstrated disruption of TNTs primarily through knockdown or inhibition of protein complexes that promote actin formation, such as M-sec (TNFaip2), Arp 2/3, and others (*Barutta et al., 2023*; *Carter et al., 2019*; *Hase et al., 2009*; *Schiller et al., 2013*). The application of TTFields has already been shown to disrupt mitosis in actively dividing cells via their effect on microtubules (MTs), using a non-pharmacologic approach. Ultimately, intracytoplasmic actin filaments are inextricably linked to microtubule dynamics (*Colin et al., 2018*), and yet the role of actin may exceed the role of MTs in cancer-relevant cellular actions (*Bijman et al., 2008*), including local invasion and metastatic potential. What is notable about TNTs is that there are sub-classes characterized by 'thin' and 'thick' variants that vary in composition, including presence and extent of microtubules. MTs are not universally expressed/present in TNTs and furthermore are highly dynamic components. A previous study from our group (*Jana et al., 2022*) using the same mesothelioma cells (MSTO-211H) reported that co-localization of different cytoskeletal elements revealed a striking difference in the nanotube structure based on geometric width. 'Thin' (<700 nm) nanotubes contained only actin, while both microtubules and vimentin intermediate filaments were localized to 'thick' (>700 nm) nanotubes. When grown in standard culture plate conditions, MSTO-211H cells formed thin TNTs that did not harbor MTs and showed a lack of tubulin expression. Within this context, our evaluation of effect of TTFields was focused on actin-based mechanisms.

Because TNTs are composed of polar actin subunits, a significant disruption of TNTs would suggest that actin is required for nanotube stability. G-actin subunits, the main component of TNTs, contain a distinct polarity. TTFields could be used to force G-actin subunits to align along the electric field instead of polymerizing. However, as there was no effect of TTFields on cell-free forms of actin, our findings suggested a more selective mechanism of TTFields. Indeed, when controlling for other parameters, the maximum suppression of TNT occurring unidirectionally vs bidirectionally may indicate that the orientation of the affected TNT component as well as its identity may play important roles in TNT formation. Thus, further studies unifying the mechanism of TTFields and the ultra-structure of TNTs are needed.

Preclinical models of TTFields have demonstrated their ability to induce cell death over time. In the current in vitro study, overall cell viability remained above 95% in both the control and treatment groups at both intensities and at all time points when 40,000 MSTO-211H cells were seeded one day before treatment started. However, when MSTO-211H were seeded at a lower density and exposed to TTFields, markedly reduced cell counts were observed at 72–96 hr of exposure, indicating that TTFields cytotoxic effect, at least in vitro, is affected by cell density. Ultimately, TTFields should be used in conjunction with other forms of cancer therapy, such as radiation therapy or chemotherapy to achieve maximum efficacy.

MPM is an ideal model for in vitro study and characterization of TNTs. It thus proved especially useful here, with additional value and background that MPM cells lines, including MSTO-211H had previously been evaluated after exposure to TTFields. Giladi et al. conducted a study on the optimal inhibitory frequencies and intensities of various cell lines exposed to TTFields. It was found that of the 30 cell lines tested, MSTO-211H was categorized as sensitive to the cytotoxic effects of TTFields (*Giladi et al., 2015*). Such a finding could explain the discrepancy in TNT formation between MSTO-211H and VAMT, suggesting that properties specific to individual cell lines could allow for resistance to TTFields treatment. While optimal inhibitory frequency/intensity of VAMT sarcomatoid MPM was

not measured in the study, its behavior under TTFields, specifically a non-significant difference in TNT formation, suggested that it is less sensitive relative to MSTO-211H.

The precise cellular mechanism(s) and identity of molecular machinery complex(es) necessary for TNT-mediated intercellular trafficking have not yet been identified. It is conceivable that the process mirrors the ones seen in other filamentous membrane-based protrusions, and it is equally conceivable that the process of TNTs may be cell type-dependent. Is actin necessary for the function of TNTs, or just the structure? Does actin polymerization correlate with TNT stability, in addition to function? Answers to these questions could provide insight into specific molecular markers involved in TNT formation as well as targeted therapy options in clinical practice. One idea stems from the role of the Arp 2/3 complex, which serves as a nucleation site for actin filaments by binding to the side of one filament and subsequently acting as a template for another filament, which is added at a 70 degree angle relative to the first filament (*Goode et al., 2001*). While Arp 2/3 has not directly been studied, the Rho GTPase protein family has been observed to localize multiple proteins, including Arp 2/3, that can then serve as potential nucleation sites for actin filaments (*Hanna et al., 2017*). Indeed, TTFields application has been shown to activate the Rho-ROCK pathway and promote reorganization of the actin cytoskeleton, which may explain our findings on TNT suppression in MSTO-211H (*Voloshin et al., 2020*).

The use of TTFields performed in vitro may provide insight into TNT biology. However, we sought to move a step beyond that by leveraging an even newer version of the technology that permits tailored treatment of TTFields in vivo to tumors in animal models. The TNTs that we visualized ex vivo were markedly shorter than their in vitro counterparts. These differences in length could be attributed to density of the tissue stroma and sectioning thickness, which reduces the likelihood of long-range cell communication via TNTs. We utilized TTFields technology to treat multiple MPM tumors, then further leveraged a spatial genomics approach to uncover the spatial geography of TTFields effect, and determine what links would exist, if any, between differentially expressed genes and our current and past findings of TNTs in vitro. The findings from spatial genomic analysis overall were highly notable for uncovering classes of immuno-oncologic response genes that were upregulated following TTFields exposure, in comparison to heat sham-treated tumors. The clinical implication for this finding is important because it is not yet established which set of cancer-directed therapies match best with TTFields, and in what sequence (prior to, during, or following each other), to produce best clinical response. Upregulation of factors such as CSF1 (macrophage colony stimulating factor-1, a cytokine responsible for macrophage production and immunoresponse), CX3CR1 (chemokine signaling), and HLA-DRB (lymphocyte trafficking and T cell receptor signaling), with concurrent modulation of the tumor microenvironment (TME) mediated by increased expression of collagens COL1A1, COL1A2, and COL3A1, may induce an inflammatory niche susceptible to cutting edge therapeutic including immune checkpoint inhibitors. Using COL1A1, a modulator of mesenchymal invasive potential (*Comba et al., 2022*) which we found to be upregulated following TTFields treatment (*Figure 5B–C*), provides support in identifying potential drivers of TTFields treatment resistance. Overall, our spatial genomic analysis suggests a possible role for combining TTFields with immunotherapy and/or selective targeted therapies in creating a more drug targeted friendly TME.

This work builds on evidence from our team dating back to 2010 demonstrating that TNTs and TNT-like protrusions can be imaged in intact tumors (*Lou et al., 2012*; *Lou et al., 2017*; *Lou et al., 2010*). It also, to our knowledge, provides the first report of spatial mapping and transcriptomic assessment of regions of heterogeneous tumors in tandem with identification of TNTs in intact tissue specimens, providing the potential identification of specific molecular and cellular markers of TNTs. We offer this new method, which we have designated Spatial Profiling of Tunneling nanoTubes (SPOTT), as a potential tool that will be useful for further discovery at the intersection of cellular biology and molecular diagnostics.

It is the downregulated set of genes that is most prominent in identifying signals that could explain why formation of TNTs in biphasic MPM was suppressed by TTFields. Numerous classes and specific genes involved in cell adhesion and motility or in epithelial-to-mesenchymal transition (EMT) were downregulated by TTFields-treated MPM, including most prominently Tenascin C (TNC) and vascular endothelial growth factor A (VEGFA). We have previously reported that Tenascin C, a modulator of cell invasive potential, is upregulated in mesothelioma cells primed in cell culture conditions conducive to TNT formation (*Ady et al., 2014*). Furthermore, transition of mesothelioma cells to EMT is strongly

associated with a sharp rise in TNT formation (*Lou et al., 2012*). We have previously reported on the intercellular transport of VEGF (*Lou et al., 2018*), a finding that implicates TNTs in other cancer-provoking processes including angiogenesis. In regards to the Arp2/3 complex, none of these genes were included in the transcriptome atlas we used for this study. RhoA and B expression were assayed, but no differential gene expression was observed. The data signals shown using spatiotemporal analysis produced an overview of a TME that was clearly reconfigured by TTFields treatment, one that has crossover with factors associated with TNT formation and function as shown in vitro. We also observed that TNT formation occurred primarily in high Ki-67 expression ROIs. Although there appears to be less reliance on TNTs for communication within dense stroma, there may be a functional association between in vivo TNTs and increased proliferation within the tumor. Further studies are needed to evaluate the functional capacity of TNTs within tumor tissue and the role of individual factors or groups involved in their formation and function.

Further, the current report further aligns TNTs with therapeutic targeting strategies that are already in widespread use clinically worldwide, FDA-approved for patients with MPM and glioblastoma, and under active investigation in clinical trials for multiple other forms of cancer. This work identifies TNTs as a potential therapeutic target for TTFields in MPM, and possibly other malignancies as well. An analogous cellular protrusion called 'tumor microtubes' (TMs) has been associated with invasion resistance to treatment with chemotherapy, radiation, and surgery in an animal model of glioblastoma (*Jung et al., 2017*; *Osswald et al., 2015*; *Osswald et al., 2016*; *Weil et al., 2017*). With increasing evidence for the role of TMs in glioblastoma biology, potential effects of TTFields on TMs may account at least in part for the significant improvements in overall survival (4.9 months) when TTFields are added to standard-of-care treatment (*Stupp et al., 2017*).

Limitations of this study include uncertainty of factors that are necessary, sufficient, and crucial to formation and maintenance of TNTs both in vitro and in vivo. In this context, it is uncertain as of yet why TNTs in the biphasic (epithelioid and sarcomatoid) MSTO-211H cell line responded effectively to TTFields treatment, but TNTs in the purely sarcomatoid cell line VAMT did not. All in vitro experiments were limited by the maximum size of the 22 mm coverslip used to culture cells for TTFields treatment; and only at this diameter could the coverslip fit into the ceramic dish for TTFields delivery. Thus, a delicate balance existed between plating too high a density of cells approaching confluency versus plating too few of cells such that growth rate was suboptimal.

In this study, we report novel cellular and molecular effects of TTFields in relation to tumor communication networks enabled by TNTs and related molecular pathways. TTFields significantly suppressed formation of TNTs in biphasic malignant mesothelioma (MSTO-211H). Spatial genomic assessment of TTFields treatment of intact mesothelioma tumors from an animal model sheds new light on gene expression alterations at the transcriptomic level that imply that TTFields may provide synergy with chemotherapy and immunotherapeutic strategies. These results position TNTs as potential therapeutic targets of TTFields and also identify the use of TTFields to remodulate the tumor microenvironment and enable a greater response to immunotherapeutic drugs.

## Materials and methods

**Key resources table**

| Reagent type (species) or resource | Designation | Source or reference | Identifiers | Additional information |
|---|---|---|---|---|
| Gene (Homo sapien) | Human fascin-1 | GenBank | HGNC:HGNC:11148 | |
| Strain, strain background (*Escherichia coli*) | BL21 DE3pLysS | Novagen | 69451 | competent cells |
| Cell line (Homo sapien) | Biphasic Mesothelioma | ATCC | CRL-2081 | MSTO-211H |
| Cell line (Homo sapien) | Sarcomatoid Mesothelioma | Authenticated | Authenticated | VAMT |
| Recombinant DNA reagent | pGV67 plasmid | This paper | GST/TEV expression vector derived from p21d (Novagen 69743) | doi: 10.1074/jbc.M111.322958 [PMID:18640983] *Nolen and Pollard, 2008* |

*Continued on next page*

*Continued*

| Reagent type (species) or resource | Designation | Source or reference | Identifiers | Additional information |
|---|---|---|---|---|
| Recombinant DNA reagent | GeoMx Mouse Cancer Transcriptome Atlas panel | NanoString Technologies, Inc. | GMX-RNA-NGS-CTA-4 | |
| Commercial Assay or kit | NucGreen Dead 488 ReadyProbes Reagent | Thermo Fisher Scientific | R37109 | |
| Commercial assay or kit | Click-iT TUNEL Alexa Fluor 488 Imaging Assay | Thermo Fisher Scientific | C10617 | |
| Software, algorithm | Zen Pro 2012 | Carl Zeiss Microscopy | Version 1.1.1 | |
| Software, algorithm | GeoMx DSP software | NanoString Technologies, Inc. | Version 2.4.0.421 | |
| Software, algorithm | SAS | SAS Viya | Version 9.4 | |
| Software, algorithm | Deseq package in R | R Foundation for Statistical Computing | Version 3.1 | |
| Software, algorithm | GraphPad Prism | GraphPad Software | Version 7.0 | |
| software, algorithm | Fiji-ImageJ software | Fiji organization | Version 2.9.0/1.53 t | |
| Other | Nunc Thermanox coverslips | Thermo Fisher Scientific | 174977 | 22 mm plastic cell-culture coverslips used with inovitro, found in "inovitro TTFields treatment" subheading in Materials and Methods |
| Other | 35 mm high wall, glass bottom dish | Ibidi | 81158 | |
| Other | MitoTracker Orange CMTMRos | Thermo Fisher Scientific | M7510 | fluorescent dye specific to mitochondria in cells, found in "Cargo and mitochondria transfer" subheading in Materials and Methods |

## Cell lines and culture

MSTO-211H cells are a biphasic MPM cell line that was purchased from the American Type Culture Collection (ATCC, Rockville, MD, USA) for use in this study. VAMT is a sarcomatoid MPM cell line that was authenticated prior to use using STR profiling. Both cell lines were grown in RPMI-1640, supplemented with 10% Fetal Bovine Serum (FBS), 1% Penicillin-Streptomycin, 1 x GlutaMAX (all from Gibco Life Technologies, Gaithersburg, MD, USA), and 0.1% Normocin anti-mycoplasma reagent (Invivogen, San Diego, CA, USA). Cells were confirmed as negative for mycoplasma infection and were maintained in a humidified incubator at 37 °C with 5% carbon dioxide. Cell viability was assayed by treating cells with NucGreen Dead 488 ReadyProbes Reagent (Invitrogen, Carlsbad, CA, USA), imaging seven random fields of view, and quantifying these fields. Apoptosis and DNA fragmentation were assayed with Click-iT TUNEL Alexa Fluor 488 Imaging Assay (Thermo Fisher Scientific, Waltham, MA, USA) according to the manufacturer's instructions.

## inovitro TTFields treatment

An inovitro device, provided by Novocure, Ltd (Haifa, Israel), was used to apply continuous bidirectional TTFields treatment to cells. One day prior to treatment with TTFields, 22 mm plastic cell-culture treated coverslips (Thermo Fisher Scientific Nunc Thermanox, Waltham, MA, USA) were placed inside sterile ceramic dishes. MSTO-211H cells (40,000) in 2 ml of growth media were plated onto the coverslips, and the dishes were placed in a base plate in a humidified incubator at 37 °C with 5% carbon dioxide overnight. To apply TTFields to the cells, the ceramic dishes were connected to an inovitro Generator Box. inovitro software controls and monitors the electrical resistance, voltage, and current in real time, while the temperature in the incubator is directly correlated with the intensity of the electric field. The temperature was set at 32 °C to deliver an intensity of 0.5 V/cm and at 26.5 °C for an intensity of 1.0 V/cm[20]. Additionally, the frequency of the electric field was set at 200 kHz for all conditions in both cell lines, barring any initial frequency testing and cell viability assessment. All intensity

values were expressed in root mean square (RMS) values to illustrate the conventional depiction of alternating current measurements in physics fields. The treated group was exposed to TTFields for 72 hr in both 0.5 V/cm and 1.0 V/cm experiments. For the 1.0 V/cm experiments, the TTFields were shut off at 72 hr, and the cells were incubated for another 24 hr to assess recovery of TNTs. Cells in the control group were not treated with TTFields and were plated as described above and placed in an incubator at 37 °C with 5% carbon dioxide for the duration of the experiment. The low-density experiments were run as described above with the exception that only 10,000 cells were plated onto a coverslip, and TTFields application followed 3 hr later.

## TNT analysis and quantification

Quantification and visual identification of TNTs were performed as described previously (*Lou et al., 2012*; *Rustom et al., 2004*; *Ady et al., 2016*; *Ady et al., 2014*; *Thayanithy et al., 2014*). Briefly, these parameters included (i) lack of adherence to the substratum of tissue culture plates, including visualization of TNTs passing over adherent cells; (ii) TNTs connecting two cells or if extending from one cell were counted if the width of the extension was estimated to be <1000 nm; and (iii) detection of a narrow base at the site of extrusion from the plasma membrane. Cellular extensions that were not clearly identified with the above parameters were excluded. Still images and time-lapse videos were analyzed using Fiji-ImageJ software. The Fiji-ImageJ Multi-point tool was used to quantify TNTs and cell number following the criteria detailed above; and the TNT index was calculated as the number of TNTs per 100 cells. The X, Y coordinate function was used to calculate the length of TNTs, using a conversion of 0.335 µm/pixel with a 20 x objective.

## Time-lapse microscopic imaging with concurrent continuous administration of TTFields using inovitro Live

An inovitro Live device, provided by Novocure, Ltd (Haifa Israel), was used to apply continuous unidirectional or bidirectional TTFields exposure to cells. One day prior to treatment, 40,000 MSTO-211H cells were plated onto a 35 mm high wall, glass bottom dish (Ibidi, Gräfelfing, Germany), and allowed to adhere overnight. For the unidirectional and bidirectional experiments, the glass bottom dish was coated with Poly-D-Lysine (Millipore Sigma, Burlington, MA) at a concentration of 1 mg/µM for 1 hr then dried for 2 hr prior to plating. The next day, an inovitro Live insert was positioned in the 35 mm dish, and placed in the microscope chamber. The plate was connected to an inovitro Live cable, and a heating element was added on top of the dish cover to minimize condensation from heat generated by TTFields. The cable was then connected to an inovitro Live Generator, and the software controlled the delivery of an electric field in either one (unidirectional) or two (bidirectional) directions at an intensity of 1.0 V/cm and either 150 or 200 kHz. Media was changed every 24 hr, during which TTFields were paused and then resumed once the cells were placed back into the incubator. The cells for the control group were plated as described above and placed in the microscope chamber at 37 °C, without TTFields, for the duration of the experiment. Seven Fields of View (FOV) were selected every 24 hr, up to 72 hr and both cell proliferation and TNT formation were quantified.

As an additional experimental arm, MSTO-211H cells were also treated with cisplatin (160 nM) and pemetrexed (24 nM) in conjunction with TTFields application using pre-treated ibidi plates. During these experiments, images were acquired for 4 hr at 2 min/frame, and this process repeated every 24 hr, up to 72 hr total. Both cell proliferation and TNT formation were subsequently quantified as described above.

Still images and time-lapse videos were taken on a Zeiss AxioObserver M1 Microscope. In order to deliver TTFields at an intensity of 1.0 V/cm, the microscope chamber temperature was set to 26.5 °C. Images were taken on a 20 X PlanApo-Chromat objective with a numerical aperture of 0.8. We used a Zeiss Axio Cam MR camera with 6.7x6.7 µm width, and spatial resolution (dx = dy) at 20 X was 0.335 µm/pixel. Images were acquired on Zen Pro 2012 software in brightfield.

## Cargo and mitochondria transfer

Cargo Transfer within TNTs was calculated using the Manual Tracking Plugin on Fiji-ImageJ. The X, Y coordinate of each cargo was recorded over time, and exported to a spreadsheet. To calculate velocity of cargo, X and Y pixel measurements were converted into microns using the scale factor 0.335 µm/pixel (20 x objective). Then, the distance formula was implemented for Xn and Yn values,

where n is any subsequent location of the cargo in relation to the first location, X1 and Y1. This process was repeated for each cargo track to calculate distance. Finally, each distance was divided by the time interval between frames. To track mitochondria, MSTO-211H cells were stained with MitoTracker Orange CMTMRos (Thermo Fisher Scientific, Waltham, MA, USA) and followed the same experimental setup and analysis as described above.

## Actin and fascin purification

Actin was purified from chicken skeletal muscle by one cycle of polymerization and depolymerization using standard protocols in the field (Spudich et al.). It was then filtered on Sephacryl S-300 resin (GE Healthcare) in G-buffer (2 mM Tris (pH 8.0), 0.2 mM ATP, 0.5 mM DTT, 0.1 mM CaCl$_2$) to obtain actin monomers, and stored at 4 °C. Human fascin-1 was expressed with an N-terminal glutathione s-transferase (GST) tag and a TEV cleavage recognition sequence from the pGV67 plasmid in BL21 DE3pLysS competent cells. Transformants were grown in 1 L of LB broth, induced at OD$_{600}$~0.6 with 0.5 mM IPTG, and shaken overnight (200 rpm, 17 °C). To purify fascin, cell pellets were resuspended in lysis buffer (50 mM Tris, pH 8.0, 500 mM NaCl, 1 mM DTT) and sonicated. Lysed cells were centrifuged (~30,000 x $g$, 4 °C) for 40 min to isolate the soluble cell components. Samples were rotated with glutathione agarose resin (pH 8.0) for 1 hr at 4 °C, washed, and eluted (50 mM Tris, pH 8.0, 100 mM NaCl, 1 mM DTT, 100 mM glutathione). Eluted fractions were incubated with TEV protease (1.6 µM) for GST tag cleavage and dialyzed into glutathione-free buffer overnight. To remove GST contaminants and TEV protease, samples were filtered through glutathione resin followed by amylose resin. Collected flow throughs were concentrated using centrifugal filters (MilliporeSigma Amicon, MWCO 30 K). Samples were frozen in liquid nitrogen and stored at –80 °C.

## Actin polymerization and bundling sedimentation assays

Actin was polymerized at 37 °C in KMEI buffer (50 mM KCl, 1 mM MgCl$_2$, 1 mM EGTA, 10 mM Imidazole pH 7.0) for 1 hr with and without 1.0 V/cm inovitro device TTFields treatment. Samples were centrifuged at 100,000 x $g$ for 30 min at 4 °C to separate filaments and monomers. Supernatant and pellet fractions were analyzed via SDS-PAGE (12% acrylamide). Gels were then stained with Coomassie Blue for 1 hr and destained for at least 6 hours (10% ethanol, 7.5% acetic acid). Band intensities were quantified via densitometry using Fiji-ImageJ. For bundling, actin (15 µM) was first polymerized for 1 hr at 37 °C in KMEI buffer. The assembled filaments were diluted to 3 µM and added to a solution with fascin (300 nM). After 1 hr with and without 1.0 V/cm TTFields treatment, samples were centrifuged at 10,000 x $g$ for 30 min at 4 °C to pellet bundled actin. SDS-PAGE and band quantification were carried out as described previously.

## Spatial genomics

Blocks of formalin-fixed paraffin-embedded (FFPE) mesothelioma tumors that were treated with sham heat or TTFields were generously provided by Novocure, Ltd for Nanostring GeoMx spatial transcriptomic analysis. In brief, eight female mice (*Mus musculus* species, strain C57BL, aged 13 weeks) were subcutaneously injected with AB1 mouse mesothelioma cells. After tumors formed, mice were treated with heat or TTFields using the inovivo device (Novocure, Ltd) for a total of 14 days: 7 days of treatment, 2 days of rest, and 7 days of additional treatment. The tumors were excised, formalin fixed and paraffin embedded, and sent to our lab. With these tumor blocks, one 5 µm section from each tumor was placed on a glass slide for Nanostring GeoMx analysis (Seattle, WA). The slide was incubated with Ki-67 antibodies and the GeoMx Mouse Cancer Transcriptome Atlas panel of 1,812 RNA probes. Regions of interest (ROIs) were chosen, and the unique DNA indexing-oligonucleotide tags were cleaved from the RNA probes within the ROIs. These tags were then sequenced and analyzed with GeoMx DSP software.

## Microscopy imaging of intact tumors from our animal model of MPM

Eight BALB/c mice were injected with murine AB mesothelioma cells, and then treated with either TTFields or heat sham (negative control).Tumors were excised and sectioned into 15 µm sections. These sections were mounted on coverslips, deparaffinized, and stained with a nuclear dye (Sytox Green 488, ThermofisherScientific, USA, S7020) and a phalloidin stain (Alexa Fluor 647, Thermo Fisher Scientific, USA, A22287) following permeabilization with 0.1% Triton-X. They were then mounted with

Prolong Gold Antifade reagent (Thermo Fisher Scientific, USA, P36930) and covered with 1.5 thickness coverslips.

Fluorescence images were acquired by using an inverted Nikon Ti-E microscope (NIKON A1R SI, Tokyo, Japan) and an MCL NanoDrive Piezo Z Drive stage (Mad City Labs Inc Wisconsin, United States) through a 60 x oil immersion objective lens (NA = 1.4, Plan Apo lambda correction collar, Tokyo, Japan). Samples were excited with 488 nm laser power set to 3.5% and a 638 nm laser with power set to 14.2%. Lasers were scanned with Galvano mirrors at a scanning speed of 0.25 with a zoom setting of 2.392 and a line average of 8. Pinhole diameter was set to 38.31 μm and emitted light was passed through a 408/488/561/640 dichroic mirror. Emitted light was then detected by DU4 GaAsP detectors with gain settings of 54 for the blue light detector and 96 for the far-red light detector. Filter cubes from the Chroma series were used for both dyes (99022 and 99023) for Sytox 488 and Alexa Fluor 647, respectively. Images were captured on a Hamamatsu FLASH 4 camera (Hamamatsu Photonics, Hamamatsu City, Japan) with voxel dimensions of 0.108 μm in XY and 0.222 μm in Z with PMT confocal detectors and using NIKON A1 Elements software (NIKON, Melville, New York, United States) for acquisition. To calculate step size of z stacks and optimal imaging resolution, Nyquist sampling was performed. Images were processed with iterative prediction advanced denoising followed by 3D automatic deconvolution with a theoretical point spread function (based on emission wavelengths of fluorophores used), automatic background subtraction, and spherical aberration correction. Z stacks were then rendered and animated using Imaris (Oxford Instruments, Beijing, China, version 5.42.03).

## Statistical Analysis

### inovitro Experiments

Due to lower sample sizes and skewed distributions of TNTs/cell, heteroscedastic t-tests were performed to assess significance in differences between TNTs/cell. Significance tests were performed on GraphPad Prism 7.0 (GraphPad Software, Inc, La Jolla, CA, USA). p-values less than 0.05 indicated statistically significant differences; and error bars were included in graphs to depict standard error.

### Bidirectional versus unidirectional inovitro experiments

The number of TNTs/cell after TTFields exposure was compared within treatment groups as a function of time using a linear mixed model to account for the repeated measures at each time point and treatment condition within each experiment. A compound symmetry correlation structure was assumed. Least squares means and standard errors are reported. Overall tests and pairwise comparisons are reported; and no adjustments for multiple comparisons were made. Data were analyzed using SAS 9.4 (Cary, NC) and p-values <0.05 were considered statistically significant.

### Spatial genomics

A Wald test was performed to assess significance in differentially expressed genes from TTFields vs heat treated mice using the Deseq package in R (R Foundation for Statistical Computing, Vienna, Australia). For each p value generated, a Benjamini-Hochberg adjusted p-value was acquired to reduce false-positive rate and reported.

### Animal use and ethical approval

This study was performed in strict accordance with the recommendations in the Guide for the Care and Use of Laboratory Animals of the National Institutes of Health. All of the animals were handled according to approved institutional animal care and use committee (IACUC) protocols (QSF-GLP-059) of Novocure. The protocol was approved by the Israeli National Committee Council for Experiments on Animal Subjects (IL-19-12-484). All surgery was performed under ketamine-xylazine anesthesia, and every effort was made to minimize suffering.

Animals specifically used were of the *Mus musculus* species (strain C57BL), female at 13 weeks, with no genetic modification, supplied by Envigo (Jerusalem, Israel, catalog number 2BALB/C26).

## Adherence to community standards

ARRIVE and ICJME guidelines were followed for this work.

## Acknowledgements

Research reported in this publication was supported by the National Center for Advancing Translational Sciences of the National Institutes of Health Award Number UL1-TR002494. The content is solely the responsibility of the authors and does not necessarily represent the official views of the National Institutes of Health. We thank the American Association for Cancer Research (AACR) for funding support through the AACR-Novocure Tumor-Treating Fields Research Award (grant number 1-60-62-LOU), as well as the Masonic Cancer Center (MCC) through the MCC spatial grant. Additional sponsors of research work in this field in the Lou Lab include: the Minnesota Ovarian Cancer Alliance (MOCA); The Randy Shaver Cancer Research and Community Fund; the Litman Family Fund for Cancer Research; the Mu Sigma Chapter of the Phi Gamma Delta Fraternity, University of Minnesota; Love Like Laurie Legacy; friends and family of Gayle Huntington; and Dick and Lynnae Koats. Emil Lou, MD, PhD was supported by a Research Scholar Grant, RSG-22-022-01-CDP, from the American Cancer Society. Laura A Sherer and Naomi Courtemanche were supported by NIH grant R01GM122787 awarded to Naomi Courtemanche.

Spatial genomics work was supported by the resources and staff at the University of Minnesota Genomics Center. Specifically, we thank Fernanda Rodriguez and Grant Barthel for their technical and logistical support regarding spatial genomics, and Colleen Forster for slide preparation. We also thank the University of Minnesota University Imaging Center (UIC) team for their resources and technical support; and in particular, we would like to thank Mary Brown, Patrick Wiley, and Mark Sanders for their imaging support and guidance. We thank Antonia Martinez-Conde, Boris Brant, Adi Haber, Eyal Dor-On, Yaara Porat, and Moshe Giladi from Novocure, Ltd for technical assistance including with time lapse experiments analyzed by our team, and by providing inovivo-treated tumor specimens for spatial transcriptomic analysis. We thank Michael Franklin, MS from the Division of Hematology, Oncology and Transplantation at the University of Minnesota for helpful suggestions and assistance in editing this manuscript. The authors apologize that not all pertinent and seminal references in the field could be discussed or cited in this article due to space limitations.

## Additional information

### Competing interests

Kerem Wainer-Katsir: Employee of Novocure Ltd. Emil Lou: has received honoraria and travel expenses for lab-based research talks, and equipment for laboratory-based research, Novocure, Ltd, 2018-2021. Parts of the work presented in this manuscript were performed in collaboration with Novocure, the company that provided the equipment (inovitro and inovitro live) used to apply TTFields. The other authors declare that no competing interests exist.

### Funding

| Funder | Grant reference number | Author |
|--------|------------------------|--------|
| American Association for Cancer Research | AACR-Novocure Tumor-Treating Fields Research Award 1-60-62-LOU | Emil Lou |
| American Cancer Society | 10.53354/pc.gr.153691 | Emil Lou |
| Minnesota Ovarian Cancer Alliance | | Emil Lou |
| Dick and Lynnae Koats | | Emil Lou |
| University of Minnesota | The Mu Sigma Chapter of the Phi Gamma Delta Fraternity | Emil Lou |

| Funder | Grant reference number | Author |
| --- | --- | --- |
| The Litman Family Fund for Cancer Research | | Emil Lou |
| The Love Like Laurie Legacy | | Emil Lou |
| The Randy Shaver Cancer Research and Community Fund | | Emil Lou |
| National Institutes of Health | R01GM122787 | Naomi Courtemanche Laura A Sherer |

The funders had no role in study design, data collection and interpretation, or the decision to submit the work for publication.

## Author contributions
Akshat Sarkari, Conceptualization, Data curation, Formal analysis, Validation, Investigation, Methodology, Writing – original draft, Writing – review and editing; Sophie Korenfeld, Phillip Wong, Formal analysis, Investigation, Writing – review and editing; Karina Deniz, Formal analysis, Investigation, Methodology, Writing – review and editing; Katherine Ladner, Data curation, Formal analysis, Supervision, Validation, Investigation, Visualization, Methodology, Writing – original draft, Writing – review and editing; Sanyukta Padmanabhan, Data curation, Formal analysis, Investigation, Writing – review and editing; Rachel I Vogel, Formal analysis, Validation, Methodology, Writing – review and editing; Laura A Sherer, Conceptualization, Data curation, Formal analysis, Validation, Investigation, Methodology, Writing – review and editing; Naomi Courtemanche, Conceptualization, Resources, Formal analysis, Supervision, Methodology, Writing – original draft, Writing – review and editing; Clifford Steer, Resources, Supervision, Visualization, Project administration, Writing – review and editing; Kerem Wainer-Katsir, Conceptualization, Resources, Data curation, Software, Formal analysis, Investigation, Methodology, Writing – original draft, Writing – review and editing; Emil Lou, Conceptualization, Resources, Data curation, Software, Formal analysis, Supervision, Funding acquisition, Investigation, Visualization, Methodology, Writing – original draft, Project administration, Writing – review and editing

## Author ORCIDs
Akshat Sarkari ![ORCID] https://orcid.org/0000-0002-6890-9021
Emil Lou ![ORCID] https://orcid.org/0000-0002-1607-1386

## Ethics
This study was performed in strict accordance with the recommendations in the Guide for the Care and Use of Laboratory Animals of the National Institutes of Health. All of the animals were handled according to approved institutional animal care and use committee (IACUC) protocols (QSF-GLP-059) of Novocure. The protocol was approved by the Israeli National Committee Council for Experiments on Animal Subjects (IL-19-12-484). All surgery was performed under ketamine-xylazine anesthesia, and every effort was made to minimize suffering. Animals specifically used were of the Mus musculus species (strain C57BL), female at 13 weeks, with no genetic modification, supplied by Envigo (Jerusalem, Israel, catalog number 2BALB/C26).

## Decision letter and Author response
Decision letter https://doi.org/10.7554/eLife.85383.sa1
Author response https://doi.org/10.7554/eLife.85383.sa2

# Additional files

## Supplementary files
• MDAR checklist

• Supplementary file 1. Materials and Methods for inovitro/inovitro Live treatment, actin and fascin purification and polymerization, and ex vivo microscopy.

## Data availability

All data generated or analysed during this study are included in the manuscript and supporting files.

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
