## [Editor Report]

This study is based on the hypothesis that tumor treating fields, a form of cancer therapy that exposes tumors to alternating electrical fields, has an effect on tunneling nanotubes, fine actin rich protrusions that connect cancer cells and allow intercellular communication, contributing to the tumor microenvironment and therapeutic resistance. This is an interesting hypothesis and may be of importance.

---

## [Decision Letter]

**Decision letter after peer review:**

Thank you for submitting your article "Treatment with Tumor-Treating Fields (TTFields) Suppresses Intercellular Tunneling Nanotube Formation in vitro and Upregulates Immuno-Oncologic Biomarkers in vivo in Malignant Mesothelioma" for consideration by *eLife*. Your article has been reviewed by 3 peer reviewers, one of whom is a member of our Board of Reviewing Editors, and the evaluation has been overseen by Wafik El-Deiry as the Senior Editor. The reviewers have opted to remain anonymous.

Tumor treating fields (TTF) therapy is an emerging modality in cancer treatment that has been FDA-approved for some cancers after promising Phase 3 studies. The mechanism of action is thought to be in part due to the impairment of spindle formation during cell division. Here the authors investigate the effects of TTF treatment on tunneling nanotube formation and immuno-oncologic biomarkers and find that TTF treatment may also modulate these processes, supporting additional and novel mechanisms of action underlying the efficacy of TTF therapy.

Essential revisions:

1) Data should be provided at additional TTF frequencies.

2) Data must be supported by an additional means, preferably imaging of TNTs.

3) Some supporting mechanisms should be provided.

*Reviewer #1 (Recommendations for the authors):*

1. There are no clear pictures illustrating the effects of TFF on TNTs in cellular or in vivo models.

2. There are no statistical significances shown in the graphical figures. This should be done using asterisks to indicate the degree of significance.

3. The biological mechanism is not clear. TTFs have no effect on actin, as shown in figure 2, but there is no tubulin control, which would help to show that the experimental parameters are suitable.

4. The RNAseq/Spatial profiling shows differentially expressed genes, but their relationship functionally to TNTs should be clearer. Can the authors stain and show the existence of TNTs in their in vivo model?

*Reviewer #2 (Recommendations for the authors):*

General Comments: The authors performed a unique investigation on the effect of TTFields on TNT formation. But there are still questions concerning the findings that they have to address below.

1. The authors only performed their experiments at 150 and 200 kHz. But TTFields are known to have biological effects between 100 and 500 kHz. Although the authors wanted to recapitulate the clinical scenarios at the approved frequencies of 150 and 200 kHz, this is a weak argument and this reviewer strongly thinks that further experimentations need to be done at 100 kHz, 300 kHz, and 400 kHz. The reason is that there are other protein dipoles that have natural frequencies outside of the 150 and 200 kHz range, and they may have unique biological effects on TNTs or other cellular phenomena.

2. Is there a difference between unidirectional and bidirectional TTFields at 150 kHz?

3. The authors tried to recapitulate the clinical scenario in mesothelioma but combining TTFields with cisplatin or cisplatin and pemetrexed. They found that cisplatin and TTFields have a synergistic effect. Interestingly, cisplatin and pemetrexed reduced TNT formation and the addition of TTFields did not reduce TNT further (Figure 3A). So, the biological mechanism is unclear. The authors should test the tubulin interference hypothesis by adding a tubulin inhibitor such as vincristine or paclitaxel.

4. It is unclear how TNT formation is related to findings from the spatial transcriptomic analysis. There is a suggestion of association but no data to indicate that the two are related.

5. In spatial transcriptomic analysis, it is unclear how are the regions selected in an unsupervised fashion for analysis. Maybe the authors should show the transcriptomic profiles between the confluent (less TNTs) and non-confluent (more TNTs) regions of the tumor.

Specific Comments:

Line 137: Change the wording "… most effective …" to "… more effect …" because the authors have not tested frequencies outside of 150-200 kHz.

Line 464-465: The sentence, "Transition of mesothelioma cells to EMT is strongly associated with a sharp rise in TNT formation", needs a reference.

Line 465-466: The sentence, "We also reported on the intracellular transport of VEGF, a finding that implicates TNTs in other cancer-provoking processes including angiogenesis", does not appear in Results and therefore needs a reference.

*Reviewer #3 (Recommendations for the authors):*

1) Excessive verbiage takes away from the central message. The manuscript should be substantially shortened, including but not limited to the abstract and discussion.

2) It is unclear why the authors restricted field delivery to the intensity of 1V/cm. Although 1V/cm is present in the entire treatment area, field intensity can reach 2-3V/cm in the tumor targets. This reviewer wonders if some of the negative data may have been due to the field intensity used being too low (see Point 3).

3) The authors provided molecular experiments on the effects of TTFields on F-actin polymerization and bundling, which showed no effects. The authors should repeat the experiment by increasing the intensity. Alternatively, TTFields may disrupt TNT through an F-actin-independent mechanism, such as the microtubules, a known target of TTFields. Although small TNTs are mostly F-actin-based structures, larger TNTs contain both F-actin and microtubules. Have the authors considered the size of the TNTs in these cell lines and/or whether microtubules are present in them?

4) The chemo + TTFields experiments are inconclusive and do not add to the conclusions. Why the number of TNTs are substantially lower at baseline in cultures treated with chemotherapy or chemotherapy + TTFields combinations? Also, the synergy between TTFields and chemotherapy in suppressing proliferation if present at all (not strong correlation) appears not due to the reduction in TNTs, which is a central argument of the study. Also, why the P+C+TTFields culture was discontinued early at 48hrs?

5) Although the gene expression analysis to identify the effects of TTFields on global gene expression was a reasonable approach, the rationale for geographic analysis was unclear. Do the authors expect that the effects of TTFields on different regions of the tumor vary? If that is the case, the data should be clearly presented to address that hypothesis rather than just a summation of the expression of target genes in the Nano string platform, e.g., were there differences in the expression of genes involved in TNT formation and proliferation in regions that have low or high Ki67 expression?

[Editors’ note: further revisions were suggested prior to acceptance, as described below.]

Thank you for resubmitting your work entitled "Treatment with Tumor-Treating Fields (TTFields) Suppresses Intercellular Tunneling Nanotube Formation in vitro and Upregulates Immuno-Oncologic Biomarkers in vivo in Malignant Mesothelioma" for further consideration by *eLife*. Your revised article has been evaluated by Wafik El-Deiry (Senior Editor) and a Reviewing Editor.

The manuscript has been improved but there are some remaining issues that need to be addressed, as outlined below:

*Reviewer #1 (Recommendations for the authors):*

The authors have responded to reviewers comments and the manuscript is improved.

The manuscript would be further improved if more statistical analysis can be added to the figures themselves. for example figure 1h would be improved using asterisks and figure 4 has not error bars or asterisks to indicate significance.

*Reviewer #3 (Recommendations for the authors):*

This is a revised manuscript from Sarkari et al. describing the effects of TTFields on nanotubes. This reviewer appreciates the effort the authors have put in to address previous critiques. However, most of the previous points are not adequately addressed as detailed below:

1) The manuscript remains excessively long, including the abstract.

2) The rebuttal for why the authors didn't think looking into the effects of TTFields on MTs in TNTs would be informative is not convincing. The authors are advancing a new observation that TTFields disrupt TNTs, which are mostly F-actin based. However, in vitro TTFields have minimal effects on cell-free F-actin polymerization. So what is the mechanism of TTFields effects on TNTs in cells? Measuring effects of TTFields on MTs, which are found to be important in certain TNT structures would be a reasonable step (as suggested by all 3 reviewers). Alternatively, the authors should provide some other plausible mechanistic insights into how TTFields disrupt TNTs in cells, rather than essentially stating that determining the mechanism is not the focus. Since this is central to their argument, some mechanistic evidence needs to be provided and should be the focus of the work.

3) The potential synergy between TTFields and chemotherapy has not been shown. The previous critique has not been addressed. Softening the language does not equal to a substantive address of the weakness of evidence. In addition, the response to the question why the P+C+TTFields culture was discontinued early at 48hrs as stated "We were unable to quantify TNTs and cells from the 72-hour time point for P+C+TTFields due to distortion of the captured image" is not acceptable. This experiment should be repeated and new non-distorted images past 72hrs should be obtained for the analysis.

4) This reviewer still questions the rationale for the geographic transcriptomics approach in studying the effects of TTFields on TNTs. The explanation and the new Figure 5 provided did not address this question. Figure 5 shows differences in gene expression associated with TTFields treatment in ROI with high ki67 and low Ki67. However, it is unclear if TTFields-treated tumors have more or less Ki67-high or Ki67-low regions compared to the sham heat treated control. Also, was there a difference in TNT formation in ki67-high vs ki67-low regions? Without clear evidence to suggest that these different regions have different ki67 which translate into differences in TNT formation with or without TTFields treatment, this experiment as presented does not add to the central hypothesis of a TTFields-TNT connection.

---

## [Author Response]

Essential revisions:Reviewer #1 (Recommendations for the authors):1. There are no clear pictures illustrating the effects of TFF on TNTs in cellular or in vivo models.

We thank the reviewer for this and other helpful feedback and suggestions. Tunneling nanotubes are finite dynamic extracellular structures, lasting in the range of several minutes to several hours from time of initial formation to disappearance. From what we now know and have observed, TNTs can disconnect from one end, especially as connected cells move apart via usual mechanisms of cell motility. In the current study, we did not visualize any different or unusual effects of TTFields on TNTs.

To address the reviewer’s comment, we have provided additional images and time-lapse microscopy videos as representative examples of what we observed in vitro with or without application of TTFields (Figure 6, Figure 5—figure supplement 1, Videos 1 and 2).

2. There are no statistical significances shown in the graphical figures. This should be done using asterisks to indicate the degree of significance.

We apologize for this issue and provide detailed indication of statistical significance as requested in the revised manuscript.

3. The biological mechanism is not clear. TTFs have no effect on actin, as shown in figure 2, but there is no tubulin control, which would help to show that the experimental parameters are suitable.

We appreciate the nuanced perspective, and the opportunity to elaborate and make clarifications in the text. In this study, we demonstrate that TTFields at a frequency and voltage used commonly in the clinical setting can suppress formation of TNTs, but not affect cell-free actin polymerization. We described the fact that, to date, effects of TTFields on tubulin-based intracellular microtubules (MTs) has been provided as the principal mechanism for response. We and others have proposed that this is not an exclusive cellular effect, and our focus for this study is TNTs in general as an effective therapeutic target for this approach. TTFields administered according to the outlined parameters suppressed TNT formation in biphasic mesothelioma cells. In this case, examination of tubulin and by extension microtubules would not be wholly informative because MTs are not universally expressed/present in TNTs and furthermore are highly dynamic components. A study published from our group in 2022 confirmed this fact, using the same mesothelioma cells (MSTO-211H) as in the current study

[PMID: 35454893; A Jana, K Ladner, E Lou, A Nain, *Cancers* April 2022]. In that study, we reported that close inspection of TNT-forming cancer cells revealed that cytoskeleton localization was dependent on the TNT width (Figure 3G), with both microtubules and intermediate filaments present significantly more often in thick TNTs (Figure 3G). This finding as also been reported by other groups [PMID: 25571977; PMID: 17142745; PMID: 24778759; PMID: 26094971; PMID: 30459350]. We used biophysical assessment (base eccentricity) to distinguish between “thick” and “thin” TNTs, with findings that agreed with the previously reported threshold for sizes of TNTs [PMID: 17142745]. Our characterization of colocalization of different cytoskeletal elements revealed a striking difference in the nanotube structure based on geometric width: thinner nanotubes contained only actin, while thicker nanotubes demonstrated localization of both microtubules and vimentin intermediate filaments. When grown in standard culture plate conditions, MSTO-211H cells formed thin TNTs that did not harbor MTs. Because we used this set of conditions in the current study, we would again expect a lack of tubulin expression. Thus, our evaluation of effect of TTFields was centered on actin-based mechanisms. We agree that the current study opens the door for further investigation to elaborate on specific molecular and cellular biological mechanisms of best response to TTFields.

To address this concern, we have added an abbreviated form of the above clarification to the second paragraph of the Discussion section.

4. The RNAseq/Spatial profiling shows differentially expressed genes, but their relationship functionally to TNTs should be clearer. Can the authors stain and show the existence of TNTs in their in vivo model?

We performed spatial transcriptomic profiling of an animal model of malignant mesothelioma treated with TTFields vs. heat sham as a negative control, to identify potential alterations in gene regulation of major oncogenic molecular pathways. We noted that several differentially expressed genes identified in this experiment corresponded with genes that we and others have previously identified as having potential association with TNTs. More in-depth studies will be needed to define their relationship to function of TNTs. We have modified related discussion in the revised manuscript accordingly. We have previously visualized TNTs in malignant mesothelioma tumors using confocal and other forms of three-dimensional imaging.

To address this and similar comments from other reviewers, we have provided z-stacked confocal fluorescence microscopy images with accompanying 3-dimensional renderings as supplementary videos (Figure 6, Figure 5—figure supplement 1, Videos 1 and 2).

Reviewer #2 (Recommendations for the authors):General Comments: The authors performed a unique investigation on the effect of TTFields on TNT formation. But there are still questions concerning the findings that they have to address below.1. The authors only performed their experiments at 150 and 200 kHz. But TTFields are known to have biological effects between 100 and 500 kHz. Although the authors wanted to recapitulate the clinical scenarios at the approved frequencies of 150 and 200 kHz, this is a weak argument and this reviewer strongly thinks that further experimentations need to be done at 100 kHz, 300 kHz, and 400 kHz. The reason is that there are other protein dipoles that have natural frequencies outside of the 150 and 200 kHz range, and they may have unique biological effects on TNTs or other cellular phenomena.

We thank the reviewer for this and other helpful feedback and suggestions. To address the above concern, we performed additional experiments in which we applied TTFields at the most extreme frequency (400 kHz) and determined that TNTs were also suppressed using these parameters (Figure 1figure supplement 4).

2. Is there a difference between unidirectional and bidirectional TTFields at 150 kHz?

Thank you for this question. To address this, we provided additional data demonstrating application of TTFields in unidirectional and bidirectional fashion at 150 kHz in the Supplementary Data section (Figure 1—figure supplement 1) of the revised manuscript.

3. The authors tried to recapitulate the clinical scenario in mesothelioma but combining TTFields with cisplatin or cisplatin and pemetrexed. They found that cisplatin and TTFields have a synergistic effect. Interestingly, cisplatin and pemetrexed reduced TNT formation and the addition of TTFields did not reduce TNT further (Figure 3A). So, the biological mechanism is unclear. The authors should test the tubulin interference hypothesis by adding a tubulin inhibitor such as vincristine or paclitaxel.

We thank the reviewer for this interesting perspective. Reviewer 1 had a similar question regarding tubulin, with our response noted above and as follows.

In our study, we demonstrated that TTFields at a frequency and voltage used commonly in the clinical setting can suppress formation of TNTs, but not affect cell-free actin polymerization. We described the fact that, to date, effect of TTFields on tubulin-based intracellular microtubules (MTs) has been provided as the principal mechanism for response; we and others have proposed that this is not an exclusive cellular effect, and our focus for this study is TNTs in general as an effective therapeutic target for this approach.

The microtubule-targeting drug paclitaxel has previously been investigated in combination with TTFields in an ovarian cancer model (Voloshin et al. PMID: 27561100). In that study, the authors detected an enhancement of efficacy using apoptosis and number of cells as reportable parameters; and they concluded that TTFields enhanced efficacy of paclitaxel in vitro. In our study, TTFields administered according to the outlined parameters suppressed TNT formation as our primary reportable outcome, in biphasic mesothelioma cells. In this case, examination of tubulin and by extension microtubules would not be wholly informative because MTs are not universally expressed/present in TNTs and furthermore are highly dynamic components. A study published from our group in 2022 confirmed this fact, using the same mesothelioma cells (MSTO-211H) as in the current study [PMID: 35454893; A Jana, K Ladner, E Lou, A Nain, *Cancers* April 2022]. In that study, we reported that close inspection of TNT-forming cancer cells revealed that cytoskeleton localization was dependent on the TNT width (Figure 3G), with both microtubules and intermediate filaments present significantly more often in thick TNTs (Figure 3G) as also reported by other groups [PMID: 25571977; PMID: 17142745; PMID: 24778759; PMID: 26094971; PMID: 30459350]. We used biophysical assessment (base eccentricity) to distinguish between “thick” and “thin” TNTs, with findings that agreed with the previously reported threshold for TNT sizes [PMID: 17142745]. Our characterization in the co-localization of different cytoskeletal elements revealed a striking difference in the nanotube structure based on geometric width: thinner nanotubes contained only actin, while thicker nanotubes demonstrated localization of both microtubules and vimentin intermediate filaments. When grown in standard culture plate conditions, MSTO-211H cells formed thin TNTs that did not harbor MTs. Because we used this set of conditions in the current study, we would again expect a lack of tubulin expression. Thus, our evaluation of the effects of TTFields was centered on actin-based mechanisms. We agree that the current study opens the door for further investigation to elaborate on specific molecular and cellular biological mechanisms of best response to TTFields.

To address this comment, we have added an abbreviated form of the above clarification to the second paragraph of the Discussion section.

4. It is unclear how TNT formation is related to findings from the spatial transcriptomic analysis. There is a suggestion of association but no data to indicate that the two are related.5. In spatial transcriptomic analysis, it is unclear how are the regions selected in an unsupervised fashion for analysis. Maybe the authors should show the transcriptomic profiles between the confluent (less TNTs) and non-confluent (more TNTs) regions of the tumor.

Thank you for these comments. As per our response to a similar question from Reviewer 1, we performed spatial transcriptomic profiling of an animal model of malignant mesothelioma treated with TTFields vs. heat sham as a negative control to identify potential alterations in gene regulation of major oncogenic molecular pathways. We noted that several differentially expressed genes identified in this experiment corresponded with genes that we and others have previously identified as having potential association with TNTs. Regions of interest (ROIs) were selected in representative sections of the proliferative outer portion of the tumor in addition to the more necrotic tumor core. We identified areas of relatively high vs. low mitotic activity, using Ki-67 staining. To address the reviewer’s concern, we provide additional analysis comparing the high vs. low Ki-67 portions of TTFields-treated and heat sham-treated tumors (Figure 5—figure supplement 1). More in-depth studies will be needed to define their relationship to TNT function based on confluency. We have modified accordingly our discussion to this point in the revised manuscript.

Specific Comments:Line 137: Change the wording "… most effective …" to "… more effect …" because the authors have not tested frequencies outside of 150-200 kHz.Line 464-465: The sentence, "Transition of mesothelioma cells to EMT is strongly associated with a sharp rise in TNT formation", needs a reference.Line 465-466: The sentence, "We also reported on the intracellular transport of VEGF, a finding that implicates TNTs in other cancer-provoking processes including angiogenesis", does not appear in Results and therefore needs a reference.

We thank the reviewer for the detailed suggestions. For the first point, we now provide additional data utilizing 400 kHz frequency (Figure 1—figure supplement 4). For the other points, we have modified the text of the revised manuscript accordingly.

Reviewer #3 (Recommendations for the authors):1) Excessive verbiage takes away from the central message. The manuscript should be substantially shortened, including but not limited to the abstract and discussion.

We thank the reviewer for this and other helpful feedback and suggestions. We have modified the text accordingly to improve readability.

2) It is unclear why the authors restricted field delivery to the intensity of 1V/cm. Although 1V/cm is present in the entire treatment area, field intensity can reach 2-3V/cm in the tumor targets. This reviewer wonders if some of the negative data may have been due to the field intensity used being too low (see Point 3).

With the inovitro and invitro Live equipment, intensity was set by the ambient temperature. The lower the temperature, the higher the intensity. To reach 2-3V/cm we would have to run the experiment in a room or incubator with the temperature set at 13^o^C (2V/cm) or close to 0^o^C (3V/cm), which is not feasible. We used the standard intensity of 1 V/cm due to feasibility, clinical relevance, and as it is the standard used for in vitro evaluation of TTFields [Reference: PMID 28518093].

3) The authors provided molecular experiments on the effects of TTFields on F-actin polymerization and bundling, which showed no effects. The authors should repeat the experiment by increasing the intensity. Alternatively, TTFields may disrupt TNT through an F-actin-independent mechanism, such as the microtubules, a known target of TTFields. Although small TNTs are mostly F-actin-based structures, larger TNTs contain both F-actin and microtubules. Have the authors considered the size of the TNTs in these cell lines and/or whether microtubules are present in them?

We thank the reviewer for this interesting perspective. Reviewers 1 and 2 had a similar question regarding tubulin and microtubules, with our response noted above and as follows. In this study, we demonstrated that TTFields at a frequency and voltage used commonly in the clinical setting can suppress formation of TNTs, but not affect cell-free actin polymerization. The effect of TTFields on tubulin based intracellular microtubules (MTs) has been provided as the principal mechanism for response. We and others have proposed that this is not an exclusive cellular effect, and our focus for this study was TNTs in general as an effective therapeutic target. TTFields administered according to the outlined parameters suppressed TNT formation in biphasic mesothelioma cells. In this case, examination of tubulin and by extension microtubules would not be entirely informative because MTs are not universally expressed/present in TNTs and furthermore are highly dynamic components. An article published from our group in 2022 confirmed this fact, using the same mesothelioma cells (MSTO-211H) [PMID: 35454893; A Jana, K Ladner, E Lou, A Nain, *Cancers* April 2022]. In it we reported that close inspection of TNT forming cancer cells revealed that cytoskeleton localization was dependent on TNT width (Figure 3G), with both microtubules and intermediate filaments present more significantly in thick TNTs (Figure 3G) as was also reported by other groups [PMID: 25571977; PMID: 17142745; PMID: 24778759; PMID: 26094971; PMID: 30459350]. We used biophysical assessment (base eccentricity) to distinguish between “thick” and “thin” TNTs, with findings that agreed with the previously reported threshold for sizes of TNTs [PMID: 17142745]. Our characterization of co-localization of different cytoskeletal elements revealed a striking difference in the nanotube structure based on geometric width: thinner nanotubes contained only actin, while thicker nanotubes demonstrated localization of both microtubules and vimentin intermediate filaments. When grown in standard culture plate conditions, MSTO-211H cells formed thin TNTs that did not harbor MTs. Because we used the same set of conditions in the current study, we would again expect a lack of tubulin expression. Thus, our evaluation of effect of TTFields was centered on actin-based mechanisms. We agree that the current study opens the door for further investigation to elaborate on specific molecular and cellular biological mechanisms that generate a greater response to TTFields.

To address this concern, we have added an abbreviated form of the above clarification to the second paragraph of the Discussion section.

4) The chemo + TTFields experiments are inconclusive and do not add to the conclusions. Why the number of TNTs are substantially lower at baseline in cultures treated with chemotherapy or chemotherapy + TTFields combinations? Also, the synergy between TTFields and chemotherapy in suppressing proliferation if present at all (not strong correlation) appears not due to the reduction in TNTs, which is a central argument of the study. Also, why the P+C+TTFields culture was discontinued early at 48hrs?

We appreciate this point and the opportunity to clarify. We postulated in the last paragraph of the Discussion section that synergy between TTFields may provide synergy with chemotherapy, and further investigation will be needed to substantiate the mechanisms. We have softened the language to reflect this point more clearly. The y-axis of Figure 3A indicates average number of TNTs for every 100 cells. TNTs may not form perfectly homogeneously in in vitro culture, thus there is natural variability in numbers reflected in the range shown. We were unable to quantify TNTs and cells from the 72-hour time point for P+C+TTFields due to distortion of the captured image.

5) Although the gene expression analysis to identify the effects of TTFields on global gene expression was a reasonable approach, the rationale for geographic analysis was unclear. Do the authors expect that the effects of TTFields on different regions of the tumor vary? If that is the case, the data should be clearly presented to address that hypothesis rather than just a summation of the expression of target genes in the Nano string platform, e.g., were there differences in the expression of genes involved in TNT formation and proliferation in regions that have low or high Ki67 expression?

Thank you for these comments. As per our response to similar questions from Reviewer 1 and 2, we performed spatial transcriptomic profiling of an animal model of malignant mesothelioma treated with TTFields vs. heat sham as a negative control to identify potential alterations in gene regulation of major oncogenic molecular pathways. We noted that several differentially expressed genes identified in this experiment corresponded with genes that we and others have previously identified as having potential association with TNTs. Regions of interest (ROIs) were selected in representative sections of the proliferative outer portion of the tumor in addition to the more necrotic tumor core. We identified areas of relatively high vs. low mitotic activity, using Ki-67 staining. To address the reviewer’s concern, we have provided additional analysis comparing the high vs. low Ki-67 portions of TTFields-treated and heat sham-treated tumors (Figure 5—figure supplement 1). We have modified the Discussion accordingly to address these points in the revised manuscript.

[Editors’ note: what follows is the authors’ response to the second round of review.]

Reviewer #1 (Recommendations for the authors):The authors have responded to reviewers comments and the manuscript is improved.The manuscript would be further improved if more statistical analysis can be added to the figures themselves. for example figure 1h would be improved using asterisks and figure 4 has not error bars or asterisks to indicate significance.Reviewer #3 (Recommendations for the authors):This is a revised manuscript from Sarkari et al. describing the effects of TTFields on nanotubes. This reviewer appreciates the effort the authors have put in to address previous critiques. However, most of the previous points are not adequately addressed as detailed below:1) The manuscript remains excessively long, including the abstract.2) The rebuttal for why the authors didn't think looking into the effects of TTFields on MTs in TNTs would be informative is not convincing. The authors are advancing a new observation that TTFields disrupt TNTs, which are mostly F-actin based. However, in vitro TTFields have minimal effects on cell-free F-actin polymerization. So what is the mechanism of TTFields effects on TNTs in cells? Measuring effects of TTFields on MTs, which are found to be important in certain TNT structures would be a reasonable step (as suggested by all 3 reviewers). Alternatively, the authors should provide some other plausible mechanistic insights into how TTFields disrupt TNTs in cells, rather than essentially stating that determining the mechanism is not the focus. Since this is central to their argument, some mechanistic evidence needs to be provided and should be the focus of the work.3) The potential synergy between TTFields and chemotherapy has not been shown. The previous critique has not been addressed. Softening the language does not equal to a substantive address of the weakness of evidence. In addition, the response to the question why the P+C+TTFields culture was discontinued early at 48hrs as stated "We were unable to quantify TNTs and cells from the 72-hour time point for P+C+TTFields due to distortion of the captured image" is not acceptable. This experiment should be repeated and new non-distorted images past 72hrs should be obtained for the analysis.4) This reviewer still questions the rationale for the geographic transcriptomics approach in studying the effects of TTFields on TNTs. The explanation and the new Figure 5 provided did not address this question. Figure 5 shows differences in gene expression associated with TTFields treatment in ROI with high ki67 and low Ki67. However, it is unclear if TTFields-treated tumors have more or less Ki67-high or Ki67-low regions compared to the sham heat treated control. Also, was there a difference in TNT formation in ki67-high vs ki67-low regions? Without clear evidence to suggest that these different regions have different ki67 which translate into differences in TNT formation with or without TTFields treatment, this experiment as presented does not add to the central hypothesis of a TTFields-TNT connection.

In the revised version, we have provided additional clarifications to the figures specified by reviewers, and also edited the text including the abstract as requested.